# Adolescent alcohol exposure promotes mechanical allodynia and alters synaptic function at inputs from the basolateral amygdala to the prelimbic cortex

J Daniel Obray[1], Erik T Wilkes[1], Mike Scofield[1,2], L Judson Chandler[1]*

[1]Department of Neuroscience, Medical University of South Carolina, Charleston, United States; [2]Department of Anesthesia and Perioperative Medicine, Medical University of South Carolina, Charleston, United States

*For correspondence:
chandj@musc.edu

Competing interest: The authors declare that no competing interests exist.

## eLife Assessment

This manuscript presents **important** information as to how adolescent alcohol exposure (AIE) alters pain behavior and relevant neurocircuits, with **convincing** data. The manuscript focuses on how AIE alters the basolateral amygdala, to the PFC (PV-interneurons), to the periaquaductal gray circuit, resulting in feed-forward inhibition. The manuscript is a detailed study of the role of alcohol exposure in regulating the circuit and reflexive pain, however, the role of the PV interneurons in mechanistically modulating this feed-forward circuit could be more strongly supported.

**Abstract** Binge drinking is common among adolescents despite mounting evidence linking it to various adverse health outcomes that include heightened pain perception. The prelimbic (PrL) cortex is vulnerable to insult from adolescent alcohol exposure and receives input from the basolateral amygdala (BLA) while sending projections to the ventrolateral periaqueductal gray (vlPAG) – two brain regions implicated in nociception. In this study, adolescent intermittent ethanol (AIE) exposure was carried out in male and female rats using a vapor inhalation procedure. Assessments of mechanical and thermal sensitivity revealed that AIE exposure-induced protracted mechanical allodynia. To investigate synaptic function at BLA inputs onto defined populations of PrL neurons, retrobeads and viral labeling were combined with optogenetics and slice electrophysiology. Recordings from retrobead labeled cells in the PrL revealed AIE reduced BLA-driven feedforward inhibition of neurons projecting from the PrL to the vlPAG, resulting in augmented excitation/inhibition (E/I) balance and increased intrinsic excitability. Consistent with this finding, recordings from virally tagged PrL parvalbumin interneurons (PVINs) demonstrated that AIE exposure reduced both E/I balance at BLA inputs onto PVINs and PVIN intrinsic excitability. These findings provide compelling evidence that AIE alters synaptic function and intrinsic excitability within a prefrontal nociceptive circuit.

## Introduction

Alcohol misuse and pain exhibit a bidirectional relationship. Due to its analgesic properties, alcohol is often used to self-medicate for pain relief (*Alford et al., 2016*; *Riley and King, 2009*). However, the dose required for this effect results in blood alcohol levels akin to binge drinking (*Neddenriep et al., 2019*; *Thompson et al., 2017*), heightening the risk of alcohol-related harm. While acute alcohol consumption can temporarily alleviate pain, chronic misuse promotes hyperalgesia (*Dina et al., 2000*;

**eLife digest** Alcohol is sometimes used as a temporary form of pain relief. However, heavy and regular consumption can have serious side effects, including altering how the brain processes pain. Over time, this may lead to more frequent and intense episodes of pain, creating a vicious cycle in which individuals drink more to counteract their heightened discomfort.

Recent studies in rodents have shown that binge drinking during adolescence also increases sensitivity to pain, with this change often persisting into adulthood. Yet, how alcohol use in teenagers impacts the parts of the brain that process pain remains poorly understood.

To investigate this question, Obray et al. compared the brains of adolescent rats that had either been exposed or not exposed to alcohol. The rats were subjected to two types of pain stimuli: mechanical pressure using the end of a metal wire and thermal pain via a heated surface. The team found that less pressure was needed for the alcohol-exposed rats to pull their paws away, suggesting they were more sensitive to pain. However, both groups of rats exhibited similar responses to the heat-related stimulus.

Next, Obray et al. explored the connections between the parts of the brain that process pain. A region of the brain known as the prefrontal cortex integrates the sensory and emotional aspects of pain by sending information to and from the amygdala and periaqueductal gray areas. Obray et al. found that inhibitory interneurons in the prefrontal cortex, which may reduce the transmission of pain, were not as well connected to the amygdala in the alcohol-exposed rats. In addition, neurons linking the prefrontal cortex to the periaqueductal gray areas were more excitable in these animals compared to the non-exposed rats.

These findings suggest that alcohol use during adolescence may make the brain more reactive to pain while impairing its ability to modulate pain signals. Understanding how early alcohol exposure alters pain sensitivity could help scientists develop strategies that disrupt the harmful cycle between alcohol use and pain. However, further studies are needed to determine whether the effects observed in this study also occur in humans.

---

*Dudek et al., 2020*; *Edwards et al., 2012*; *Jochum et al., 2010*; *Julian et al., 2019*; *You et al., 2020*). Notably, frequent acute pain and pain interference are associated with an elevated risk of being diagnosed with an alcohol use disorder (AUD) (*Barry et al., 2013*; *Edlund et al., 2013*; *McDermott et al., 2018*). This dynamic extends to adolescents where untreated pain correlates with earlier initiation of alcohol use (*Chau and Chau, 2023*), and alcohol consumption is associated with heightened ongoing and future pain (*Hestbaek et al., 2006*; *Horn-Hofmann et al., 2018*; *Pascale et al., 2022*). Recent preclinical studies in rodents further indicate that adolescent intermittent ethanol (AIE) exposure induces persistent hyperalgesia spanning into adulthood (*Bertagna et al., 2024*; *Kelley et al., 2024*; *Khan et al., 2023*; *Secci et al., 2024*). Given the association between pain and AUD, understanding the impact of adolescent alcohol use on nociceptive circuits is crucial for improving AUD treatments.

Nociception involves multimodal processing across a distributed brain network. Within this network, the medial prefrontal cortex (mPFC) is a key node for evaluating and responding to pain (*Bastuji et al., 2016*; *Garcia-Larrea and Peyron, 2013*; *Ong et al., 2019*). The prelimbic (PrL) subregion of the mPFC receives inputs from the basolateral amygdala (BLA) (*Cunningham et al., 2002*; *Gabbott et al., 2006*; *Krettek and Price, 1977*) and sends projections to the ventrolateral periaqueductal gray (vlPAG) (*An et al., 1998*; *Floyd et al., 2000*), both regions involved in nociception. Within this circuit, glutamatergic inputs from the BLA drive parvalbumin interneuron (PVIN)-mediated feedforward inhibition of pyramidal neurons projecting from the PrL to the vlPAG (PrL$^{PAG}$ neurons) to promote nociception (*Cheriyan et al., 2016*; *Dilgen et al., 2013*; *Gadotti et al., 2019*; *Huang et al., 2019*; *McGarry and Carter, 2016*). Of note, activation of PrL PVINs has been shown to be pronociceptive (*Yin et al., 2020*; *Zhang et al., 2015*), while activation of PrL$^{PAG}$ neurons generally exhibits antinociceptive effects (*Drake et al., 2021*; *Gao et al., 2023*; *Huang et al., 2019*; *Yin et al., 2020*), although this finding has not been universally observed (*Fan et al., 2018*). These findings highlight the involvement of a BLA–PrL–vlPAG circuit in modulating nociception.

Adolescence marks a critical period of continuing development of the mPFC. This period is characterized by heightened plasticity rendering this region particularly vulnerable to environmental

insults, including repeated binge alcohol consumption (*Crews et al., 2019*; *Spear, 2000*; *Spear, 2018*). Notably, this developmental phase is accompanied by significant alterations in excitatory and inhibitory neurotransmission within the mPFC. Excitatory changes include pruning of both synapses and dendritic spines (*Drzewiecki et al., 2016*; *Koss et al., 2014*; *Mallya et al., 2019*; *Petanjek et al., 2011*; *Rakic et al., 1994*), maturation of AMPA and NMDA receptor trafficking (*Flores-Barrera et al., 2014*; *Miller et al., 1990*; *Murphy et al., 2012*), and increased innervation from the BLA (*Cunningham et al., 2002*; *Cunningham et al., 2008*). Concurrently, there is a substantial increase in excitatory input onto PVINs (*Caballero et al., 2014*; *Cunningham et al., 2008*), leading to a shift toward greater inhibition in the excitatory/inhibitory (E/I) balance at pyramidal neurons (*Caballero et al., 2021*; *Cass et al., 2014*; *Klune et al., 2021*). Preclinical rodent models have demonstrated that AIE exposure disrupts the normative developmental trajectory of the mPFC, inducing persistent alterations in intrinsic excitability and synaptic function. Specifically, adolescent alcohol exposure leads to decreased intrinsic excitability of PVINs (*Trantham-Davidson et al., 2017*), reduced excitatory input onto PVINs (*Trantham-Davidson et al., 2017*), diminished inhibition at pyramidal neurons (*Centanni et al., 2017*), and augmented intrinsic excitability of pyramidal neurons in the mPFC (*Galaj et al., 2020*; *Salling et al., 2018*). These findings underscore the sensitivity of mPFC circuitry to long-lasting AIE-induced changes. Building upon this understanding, the present study investigated how AIE exposure, in conjunction with a carrageen-induced inflammatory paw pain challenge, alters synaptic function at BLA inputs onto PVINs and PrL^PAG neurons.

## Results

The procedure for adolescent alcohol exposure used in this study is a well-characterized model designed to simulate the effects of repeated episodes of binge-like alcohol exposure. Rats were subjected to eight intermittent cycles of ethanol vapor from PD 28 to PD 54. Behavioral intoxication and blood ethanol concentrations (BECs) were assessed at the end of each cycle. The average behavioral intoxication score using the 5-point rating scale was 2.2 ± 0.1 for male rats in the AIE-saline treatment condition, 2.2 ± 0.1 for male rats in the AIE-carrageenan treatment condition, 2.2 ± 0.1 for female rats in the AIE-saline treatment condition, and 2.1 ± 0.1 for female rats in the AIE-carrageenan treatment condition, which represents a moderate level of intoxication. The corresponding BEC values were 196.4 ± 34.8 for male rats in the AIE-saline treatment condition, 240.0 ± 33.8 for male rats in the AIE-carrageenan treatment condition, 184.8 ± 38.8 for female rats in the AIE-saline treatment condition, and 177.7 ± 31.2 for female rats in the AIE-carrageenan treatment condition. Male rats had significantly higher intoxication scores than female rats (Wilcoxon rank-sum test: $z$ = 1.967, p = 0.0492). There was no difference in average intoxication scores between rats assigned to the saline and carrageenan pain conditions (Wilcoxon rank-sum test: $z$ = 0.450, p = 0.6529). There was no difference in BEC level between male and female rats (Wilcoxon rank-sum test: $z$ = 1.365, p = 0.1724) or for rats assigned to the saline and carrageenan pain conditions (Wilcoxon rank-sum test: $z$ = –0.955, p = 0.3395). Average intoxication scores and BEC levels were positively correlated ($r_s$ = 0.499, p = 0.0001).

### AIE exposure augmented mechanical sensitivity

The first set of studies evaluated the impact of AIE on mechanical and thermal sensitivity from adolescence to early adulthood. Weekly assessments were conducted using electronic Von Frey and Hargreaves apparatuses. The initial evaluation was performed at PD 24 prior to the first cycle of ethanol vapor exposure, and the final assessment at PD 80, which was approximately 4 weeks after the last cycle of exposure.

For mechanical sensitivity, analysis of data from the electronic Von Frey test revealed that AIE significantly reduced paw withdrawal threshold, indicating increased sensitivity to mechanical touch (main effect of AIE: $F_{(1,115)}$ = 12.81, p = 0.0005, partial $\eta^2$ = 0.1002 [0.0203, 0.2105]; *Figure 1A, B*). Additionally, female rats exhibited greater sensitivity to mechanical touch compared to male rats (main effect of sex: $F_{(1,115)}$ = 5.98, p = 0.0160, partial $\eta^2$ = 0.0494 [0.0014, 0.1434]). There was no significant interaction between AIE and sex (AIE × sex interaction: $F_{(1,115)}$ = 1.02, p = 0.3147). Mechanical touch sensitivity decreased with age (main effect of age: $F_{(8,920)}$ = 75.08, p = 0.0000, partial $\eta^2$ = 0.3950 [0.3444, 0.4328]), with no significant interactions between age and AIE or sex (AIE × age interaction: $F_{(8,920)}$ = 0.70, p = 0.6327; sex × age interaction: $F_{(8,920)}$ = 1.50, p = 0.1829; AIE × sex × age interaction:

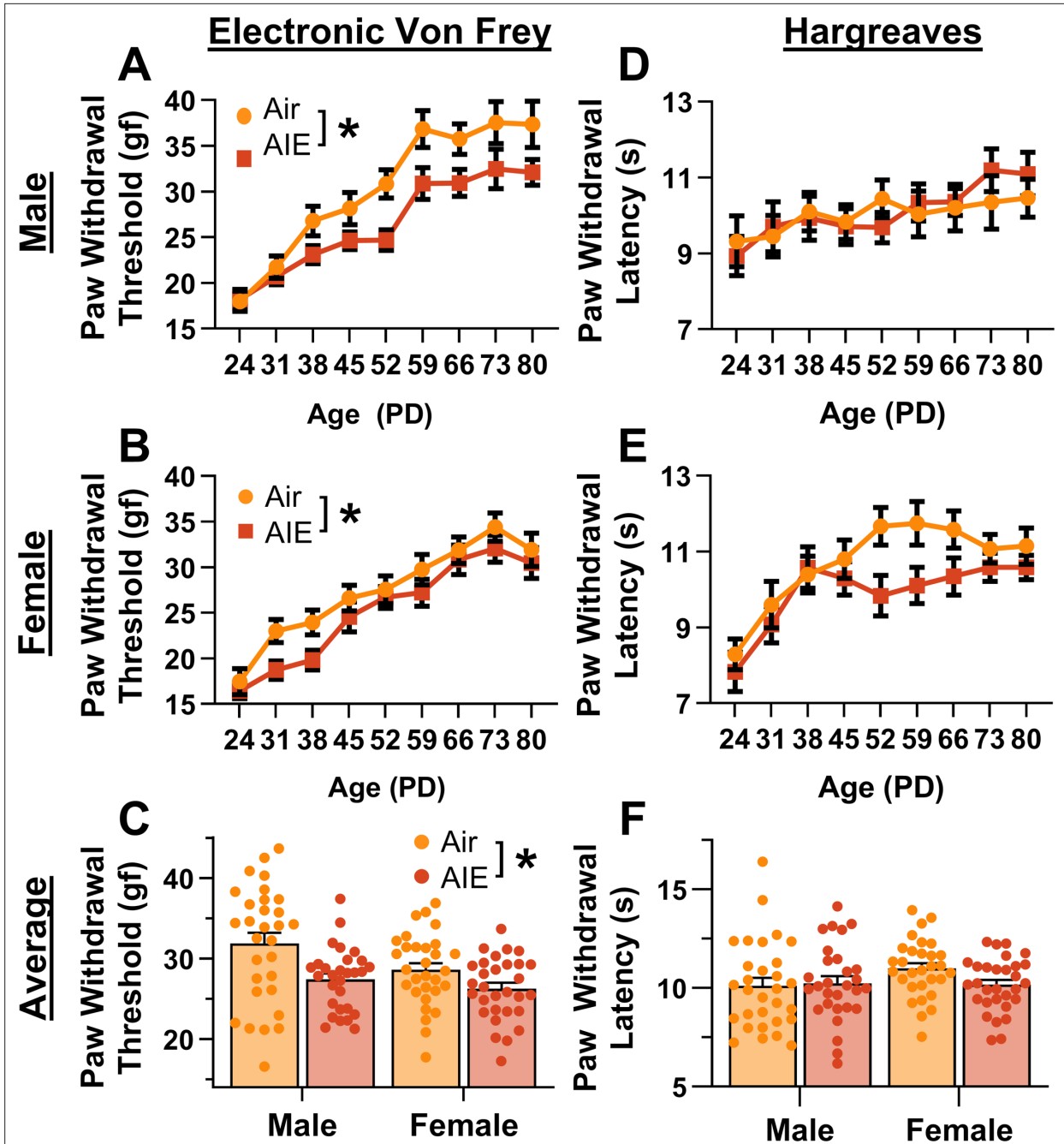

**Figure 1.** Mechanical and thermal sensitivity of rats during adolescence and early adulthood. Depiction of the effect of adolescent intermittent ethanol (AIE) exposure on mechanical sensitivity across adolescence and into early adulthood in male (**A**) and female (**B**) rats. (**C**) AIE exposure significantly reduced the average (PD 31–PD 80) electronic Von Frey (eVF) withdrawal threshold, indicating increased mechanical touch sensitivity. Depiction of the effect of AIE exposure on thermal sensitivity during adolescence and early adulthood in male (**D**) and female (**E**) rats. (**F**) AIE exposure did not significantly alter the average (PD 31–PD 80) Hargreaves test withdrawal latency, indicating no change in thermal sensitivity. Data represent the mean ± SEM. Source data for all panels is included in *Figure 1—source data 1*. Data were analyzed using ANOVA, with exposure (AIR vs. AIE), sex, and age (when applicable) as factors. * indicates a significant difference between the related conditions; $p < 0.05$; $n$ = 29–30 rats/group.

The online version of this article includes the following source data for figure 1:

**Source data 1.** Numerical data for mechanical and thermal sensitivity of rats during adolescence.

$F_{(8,920)}$ = 1.10, p = 0.3602). The data were also averaged for each rat starting after the first cycle of AIE (PD 31–PD 80). Consistent with the full dataset analysis, the averaged data showed that AIE increased sensitivity to mechanical touch (main effect of AIE: $F_{(1,115)}$ = 13.33, p = 0.0004, partial $\eta^2$ = 0.1039 [0.0221, 0.2149]; *Figure 1C*) and female rats were more sensitive to mechanical touch than male rats (main effect of sex: $F_{(1,115)}$ = 5.63, p = 0.0193, partial $\eta^2$ = 0.0467 [0.0008, 0.1393]). There was no significant interaction between AIE and sex (AIE × sex interaction: $F_{(1,115)}$ = 1.28, p = 0.2608).

For thermal sensitivity, analysis of data from the Hargreaves test indicated no significant effect of AIE on paw withdrawal latency (main effect of AIE: $F_{(1,115)}$ = 1.21, p = 0.2738; *Figure 1D, E*). Additionally, there were no significant sex differences or interactions with AIE (main effect of sex: $F_{(1,115)}$ = 0.59, p = 0.4446; AIE × sex interaction: $F_{(1,115)}$ = 1.85, p = 0.1767). Thermal sensitivity was found to decrease with age (main effect of age: $F_{(8,920)}$ = 11.45, p = 0.0000, partial $\eta^2$ = 0.0906 [0.0520, 0.1197]). Furthermore, no significant interactions between age and AIE or sex were observed (AIE × age interaction: $F_{(8,920)}$ = 1.11, p = 0.3567; sex × age interaction: $F_{(8,920)}$ = 2.02, p = 0.0546; AIE × sex × age interaction: $F_{(8,920)}$ = 0.70, p = 0.6626). The data were also analyzed as the average paw withdrawal latency for each rat beginning after the first cycle of AIE (PD 31–PD 80). Consistent with the results from the full dataset, analysis of the averaged data showed no significant effect of AIE, sex, or interaction between the two on thermal sensitivity (main effect of AIE: $F_{(1,115)}$ = 1.09, p = 0.2977; main effect of sex: $F_{(1,115)}$ = 1.57, p = 0.2128; AIE × sex interaction: $F_{(1,115)}$ = 2.20, p = 0.1408; *Figure 1F*).

## Carrageenan-induced inflammatory paw pain was unaltered by AIE exposure

As AIE was found to increase baseline mechanical sensitivity, the subsequent studies assessed its impact on carrageenan-induced hyperalgesia in adult rats. Carrageenan is a well-known proinflammatory agent that induces edema and transient hyperalgesia in the carrageenan-induced inflammatory paw pain model (*Benitz and Hall, 1959*; *Neves et al., 2020*; *Vazquez et al., 2015*; *Winter et al., 1962*; *Yang and Tsaur, 2023*).

Prior to administering carrageenan or saline into the hindpaw, baseline mechanical and thermal sensitivity was assessed using the electronic Von Frey and Hargreaves tests, respectively. Analysis revealed that AIE-driven reductions in paw withdrawal threshold persisted for more than 8 weeks after discontinuing ethanol vapor exposure (main effect of AIE: $F_{(1,109)}$ = 5.50, p = 0.0209, partial $\eta^2$ = 0.0480 [0.0006, 0.1441]; *Figure 2A*). In addition, female rats continued to display greater mechanical sensitivity than males, with no significant interactions between AIE and sex (main effect of sex: $F_{(1,109)}$ = 10.66, p = 0.0015, partial $\eta^2$ = 0.0891 [0.0139, 0.1998]; AIE × sex interaction: $F_{(1,109)}$ = 0.93, p = 0.3367). Baseline analysis of the thermal sensitivity data showed no significant effects of AIE, sex, or an interaction between AIE and sex on paw withdrawal latency (main effect of AIE: $F_{(1,109)}$ = 1.69, p = 0.1968; main effect of sex: $F_{(1,109)}$ = 1.27, p = 0.2616; AIE × sex interaction: $F_{(1,109)}$ = 0.38, p = 0.5384; *Figure 2B*).

Following carrageenan or saline administration, mechanical and thermal sensitivity were evaluated, and data were averaged across assessments occurring at three timepoints – 2-, 6-, and 24-hr post-injection. The averaged data were expressed as a percentage of each rat's pre-injection baseline score. Examination of paw withdrawal threshold revealed a carrageenan-induced increase in mechanical sensitivity (main effect of carrageenan: $F_{(1,105)}$ = 116.41, p = 0.0000, partial $\eta^2$ = 0.5258 [0.3934, 0.6201]; *Figure 2C–F*). Analysis further indicated that female rats displayed a greater increase in sensitivity than males across both the saline and carrageenan treatments, with no significant effects of AIE, or interactions between AIE, sex, and carrageenan (main effect of sex: $F_{(1,105)}$ = 6.22, p = 0.0142, partial $\eta^2$ = 0.0559 [0.0020, 0.1578]; main effect of AIE: $F_{(1,105)}$ = 0.93, p = 0.3366; AIE × sex interaction: $F_{(1,105)}$ = 0.03, p = 0.8706; AIE × carrageenan interaction: $F_{(1,105)}$ = 0.59, p = 0.4441; sex × carrageenan interaction: $F_{(1,105)}$ = 1.72, p = 0.1925; AIE × sex × carrageenan interaction: $F_{(1,105)}$ = 1.18, p = 0.2790). Similarly, examination of paw withdrawal latency revealed a carrageenan-induced increase in thermal sensitivity (main effect of carrageenan: $F_{(1,105)}$ = 132.70, p = 0.0000, partial $\eta^2$ = 0.5583 [0.4308, 0.6470]; *Figure 2G–J*). No significant effects of AIE, sex, or interactions between AIE, sex, and carrageenan were observed (main effect of AIE: $F_{(1,105)}$ = 0.00, p = 0.9625; main effect of sex: $F_{(1,105)}$ = 0.00, p = 0.9510; AIE × sex interaction: $F_{(1,105)}$ = 0.23, p = 0.6307; AIE × carrageenan interaction: $F_{(1,105)}$ = 0.01, p = 0.9352; sex × carrageenan interaction: $F_{(1,105)}$ = 0.14, p = 0.7120; AIE × sex × carrageenan interaction: $F_{(1,105)}$ = 3.85, p = 0.0524).

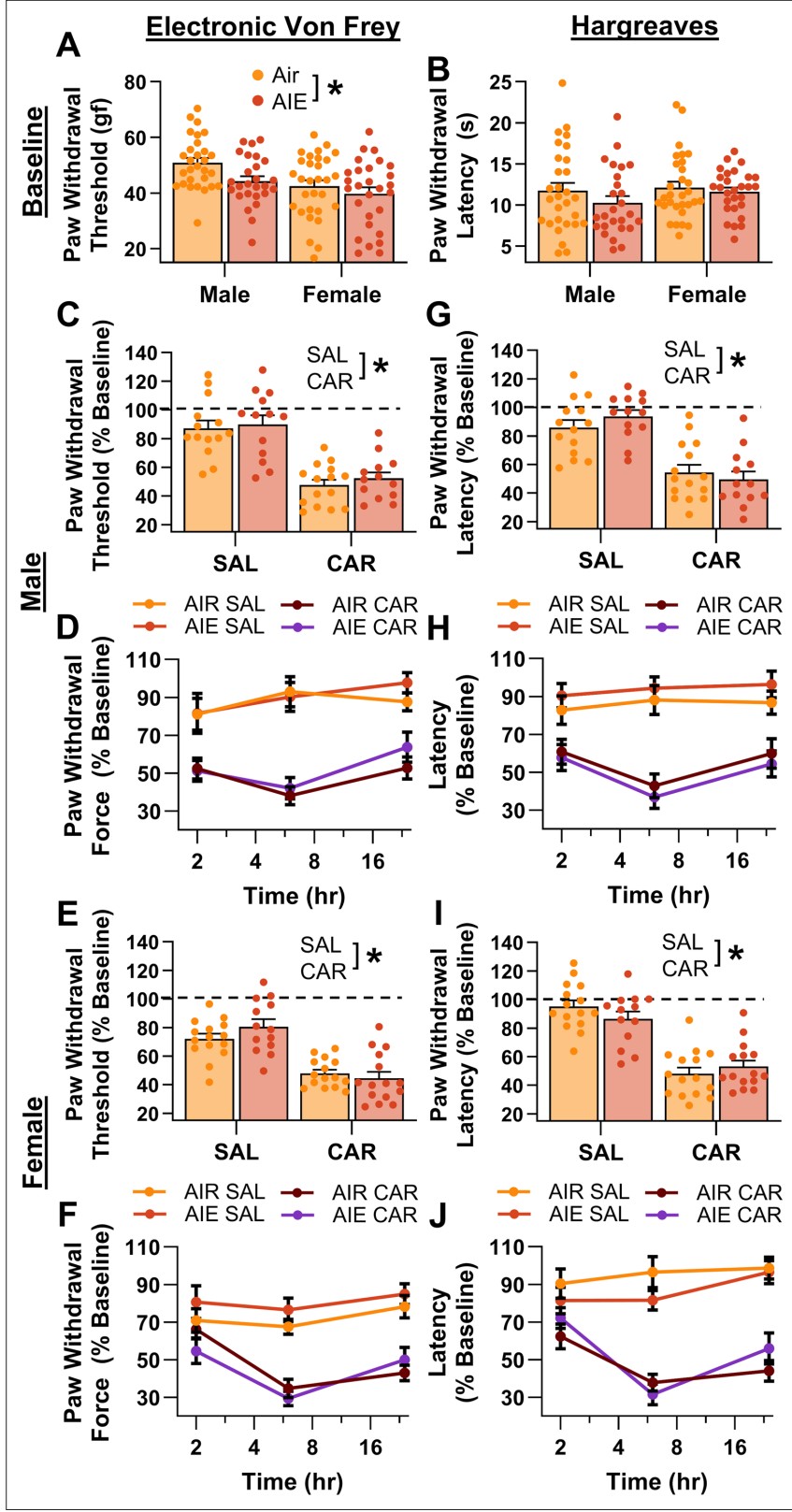

**Figure 2.** Mechanical and thermal sensitivity of rats in response to a carrageenan paw pain challenge. (**A**) Baseline mechanical touch sensitivity was greater in adolescent intermittent ethanol (AIE) exposed rats than in Air control rats. (**B**) There was no difference in baseline thermal sensitivity between AIE exposed and Air control rats. Male (**C, D**) and female (**E, F**) rats injected with carrageenan (CAR) in the right hindpaw displayed mechanical

*Figure 2 continued on next page*

*Figure 2 continued*

hypersensitivity. This hypersensitivity was not significantly altered by AIE exposure. Average paw withdrawal threshold combining all post-injection timepoints for males (**C**) and females (**E**). Paw withdrawal threshold expressed as a percentage of the baseline threshold at 2, 6, and 24 hr post-injection for males (**D**) and females (**F**). Similarly, male (**G, H**) and female (**I, J**) rats injected with CAR into the right hindpaw displayed thermal hyperalgesia, with no effect of AIE exposure on this hyperalgesia. Average paw withdrawal latency across all post-injection timepoints for male (**G**) and female (**I**) rats. Paw withdrawal latency as a percentage of the baseline latency at 2, 6, and 24 hr post-injection for male (**H**) and female (**J**) rats. Data represent the mean ± SEM. Source data for all panels is included in *Figure 2—source data 1*. Data were analyzed using ANOVA, with exposure (AIR vs. AIE), treatment (CAR vs. SAL), and sex as factors. * indicates a significant difference between the related conditions; $p <$ 0.05; $n$ = 13–15 rats/group.

The online version of this article includes the following source data for figure 2:

**Source data 1.** Numerical data for mechanical and thermal sensitivity of rats in response to a carrageenan paw pain challenge.

## AIE exposure and carrageenan enhanced the intrinsic excitability of PrL^PAG neurons

Subsequently, the impact of AIE exposure- and carrageenan-induced hyperalgesia on PrL^PAG neuron intrinsic excitability was examined through current-clamp recordings of current evoked firing obtained from green retrobead labeled cells in the PrL cortex. Labeling with green retrobeads indicated that the neuron projected ipsilateral from the left hemisphere of the PrL cortex to the vlPAG (*Figure 3*). For each electrophysiological experiment, values are reported per animal and reflect the average value of 1–5 neurons recorded from each rat. *Table 1* contains a summary of the biophysical properties of

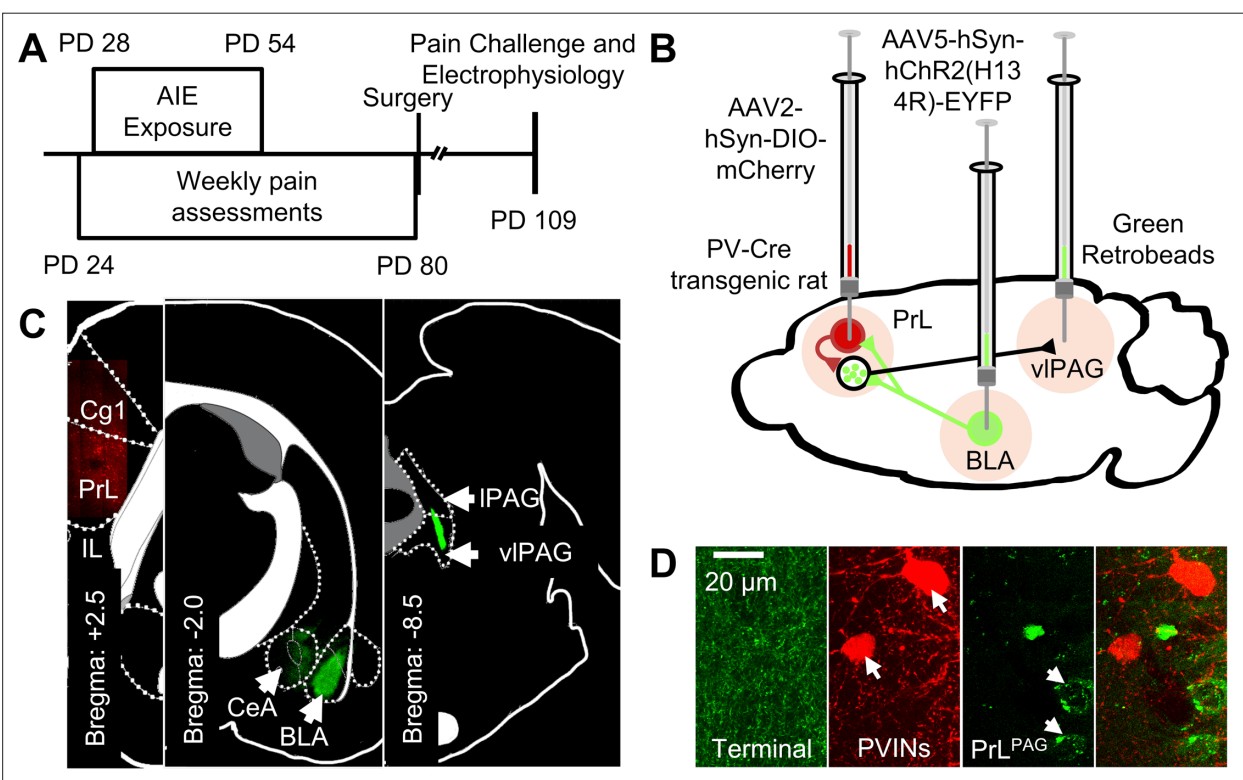

**Figure 3.** Experimental approach. (**A**) Experimental timeline displaying the age of animals at each phase in the study. (**B**) Diagram showing the viral and retrobead labeling approach used to identify and manipulate specific neuronal populations in the prelimbic (PrL) cortex. (**C**) Representative images showing injection sites in the PrL (left panel, AAV2-hSyn-DIO-mCherry), basolateral amygdala (BLA, center panel, AAV5-hSyn-hChR2(H134R)-EYFP), and ventrolateral periaqueductal gray (vlPAG, right panel, green retrobeads). (**D**) Representative images from the PrL cortex (from left to right) of BLA terminals, mCherry-tagged parvalbumin interneurons (PVINs), and green retrobead labeled PrL^PAG neurons.

**Table 1.** Biophysical properties of PrL$^{PAG}$ neurons across treatment condition and sex.

| Condition | Sex | V$_{rest}$ (mV) | R$_{input}$ (MΩ) |
|---|---|---|---|
| AIR: SAL | Male | –66.6 ± 0.9 | 75.3 ± 3.5 |
| | Female | –65.0 ± 1.7 | 76.3 ± 2.4 |
| AIR: CAR | Male | –66.2 ± 1.4 | 83.2 ± 5.4 |
| | Female | –65.2 ± 1.5 | 83.9 ± 4.1 |
| AIE: SAL | Male | –65.0 ± 1.4 | 86.9 ± 7.8 |
| | Female | –65.6 ± 1.5 | 76.6 ± 2.9 |
| AIE: CAR | Male | –65.6 ± 1.0 | 88.6 ± 4.3 |
| | Female | –66.5 ± 1.2 | 83.4 ± 6.0 |

the recorded PrL$^{PAG}$ neurons. No significant differences between treatment conditions or sex were observed for these properties.

Analysis of the firing data indicated that both AIE and carrageenan enhanced intrinsic excitability (main effect of AIE: $F_{(1,72)}$ = 7.24, p = 0.0089, partial $\eta^2$=0.0914 [0.0059, 0.2292]; main effect of carrageenan: $F_{(1,72)}$ = 4.07, p = 0.0474, partial $\eta^2$=0.0535 [0.0000, 0.1776]; *Figure 4A–G*). Moreover, the effect of AIE on intrinsic excitability became more pronounced with increasing current step size (AIE × current step interaction: $F_{(20,1440)}$ = 3.82, p = 0.0117, partial $\eta^2$ = 0.0503 [0.0194, 0.0604]). Although the number of action potentials (APs) fired increased alongside the amount of injected current (main effect of current step: $F_{(20,1440)}$ = 230.22, p = 0.0000, partial $\eta^2$ = 0.7618 [0.7404, 0.7747]), no other significant effects of sex or interactions between AIE, carrageenan, sex, or current step were observed (main effect of sex: $F_{(1,72)}$ = 1.78, p = 0.1861; AIE × carrageenan interaction: $F_{(1,72)}$ = 0.00, p = 0.9760; AIE × sex interaction: $F_{(1,72)}$ = 0.17, p = 0.6823; sex × carrageenan interaction: $F_{(1,72)}$ = 0.00, p = 0.9686; AIE × sex × carrageenan interaction: $F_{(1,72)}$ = 0.46, p = 0.4985; sex × current step interaction: $F_{(20,1440)}$ = 1.27, p = 0.2851; carrageenan × current step interaction: $F_{(20,1440)}$ = 1.13, p = 0.3377; AIE × sex × current step interaction: $F_{(20,1440)}$ = 0.66, p = 0.5694; AIE × carrageenan × current step interaction: $F_{(20,1440)}$ = 2.46, p = 0.0656; sex × carrageenan × current step interaction: $F_{(20,1440)}$ = 0.20, p = 0.8924; AIE × sex × carrageenan × current step interaction: $F_{(20,1440)}$ = 1.13, p = 0.3383).

To better understand the observed changes in intrinsic excitability, additional analyses were performed to assess the effects of AIE, carrageenan, and sex on AP threshold, rheobase, voltage sag, and afterhyperpolarization in PrL$^{PAG}$ neurons. This analysis revealed that carrageenan reduced the AP threshold of PrL$^{PAG}$ neurons, with the largest reduction observed in AIE exposed rats (main effect of carrageenan: $F_{(1,72)}$ = 7.01, p = 0.0100, partial $\eta^2$ = 0.0887 [0.0051, 0.2258]; AIE × carrageenan interaction: $F_{(1,72)}$ = 5.52, p = 0.0215, partial $\eta^2$ = 0.0713 [0.0007, 0.2029]; *Figure 4H, I*). No additional effects of treatment condition or sex on the AP threshold were found (main effect of sex: $F_{(1,72)}$ = 3.23, p = 0.0764; main effect of AIE: $F_{(1,72)}$ = 3.48, p = 0.0662; AIE × sex interaction: $F_{(1,72)}$ = 0.17, p = 0.6794; sex × carrageenan interaction: $F_{(1,72)}$ = 0.15, p = 0.7004; AIE × sex × carrageenan interaction: $F_{(1,72)}$ = 1.31, p = 0.2561). Despite the observed change in AP threshold, there were no significant effects of treatment condition or sex on rheobase in these neurons (main effect of AIE: $F_{(1,72)}$ = 0.71, p = 0.4014; main effect of carrageenan: $F_{(1,72)}$ = 0.71, p = 0.4014; main effect of sex: $F_{(1,72)}$ = 0.13, p = 0.7186; AIE × carrageenan interaction: $F_{(1,72)}$ = 1.76, p = 0.1889; AIE × sex interaction: $F_{(1,72)}$ = 0.01, p = 0.9044; sex × carrageenan interaction: $F_{(1,72)}$ = 0.71, p = 0.4014; AIE × sex × carrageenan interaction: $F_{(1,72)}$ = 1.18, p = 0.2814; *Figure 4J, K*). Hyperpolarization-activated cation current ($I_h$) linked voltage sag was reduced by carrageenan (main effect of carrageenan: $F_{(1,72)}$ = 5.03, p = 0.0281, partial $\eta^2$ = 0.0652 [0.0000, 0.1946]; *Figure 4L, M*), with the largest reduction occurring in male rats (sex × carrageenan interaction: $F_{(1,72)}$ = 5.20, p = 0.0256, partial $\eta^2$ = 0.0673 [0.0000, 0.1975]). As a result, male carrageenan treated rats had smaller $I_h$-induced hyperpolarization sag than female rats (main effect of sex: $F_{(1,72)}$ = 20.92, p < 0.0001, partial $\eta^2$ = 0.2252 [0.0758, 0.3746]). Afterhyperpolarization, in contrast, was reduced by AIE exposure (main effect of AIE: $F_{(1,72)}$ = 53.33, p < 0.0001, partial $\eta^2$ = 0.4255 [0.2518, 0.5535]; *Figure 4N, O*) with the largest reduction occurring in females (AIE × sex interaction: $F_{(1,72)}$

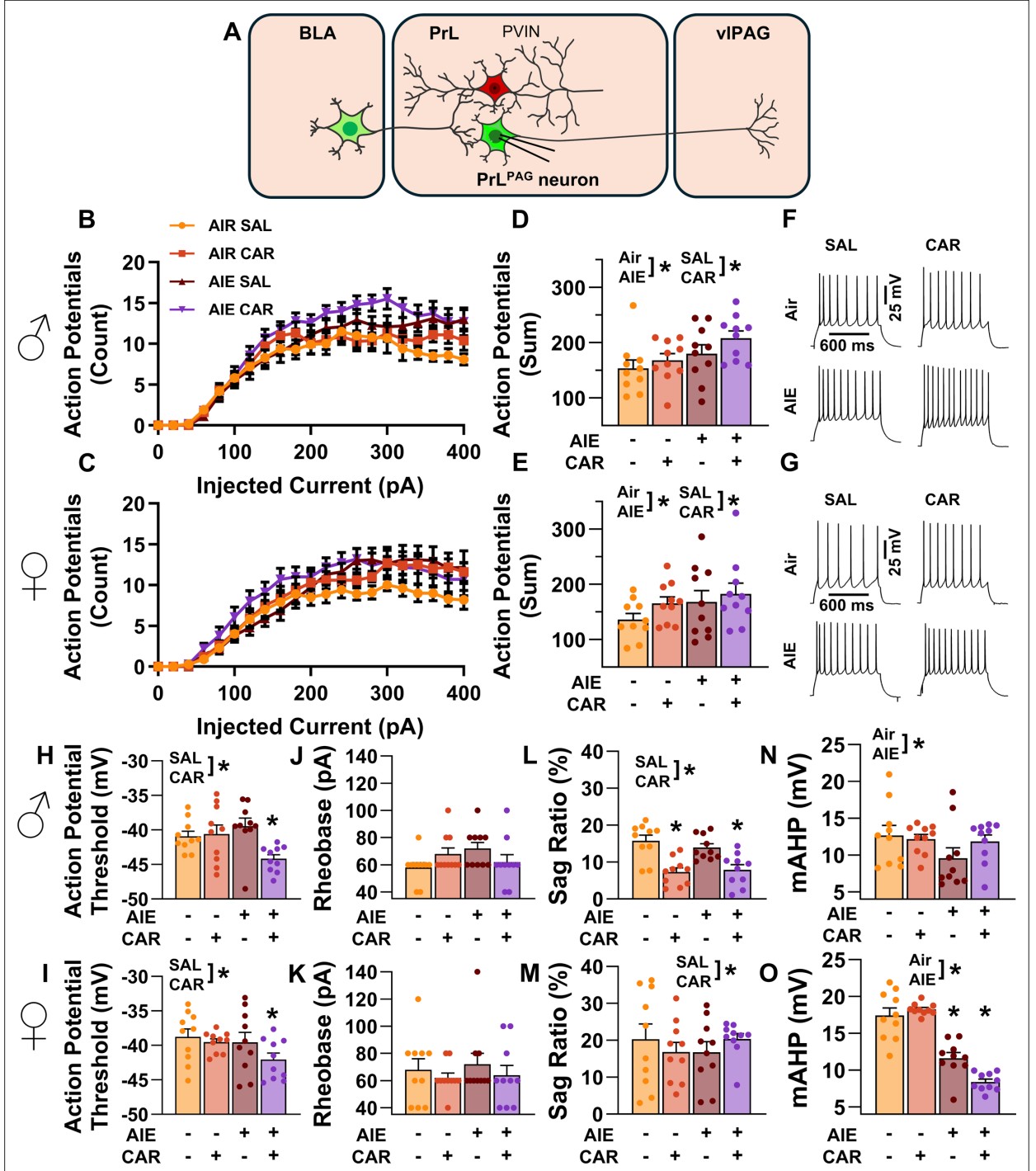

**Figure 4.** Intrinsic excitability of pyramidal neurons projecting from the prelimbic cortex to the ventrolateral periaqueductal gray (PrL$^{PAG}$).
(**A**) Electrophysiological recordings were obtained from PrL$^{PAG}$ neurons in the PrL cortex. Depiction of the relationship between injected current and action potential firing in male (**B**) and female (**C**) rats across treatment conditions. The cumulative number of action potentials fired across all current steps was increased by both adolescent intermittent ethanol (AIE) exposure and a carrageenan paw pain challenge (CAR) in male (**D**) and female (**E**) rats. Representative traces showing action potential spiking across treatment conditions in male (**F**) and female (**G**) rats. The action potential threshold of PrL$^{PAG}$ neurons was reduced in AIE exposed, CAR treated male (**H**) and female (**I**) rats. The rheobase of PrL$^{PAG}$ neurons was unaltered by AIE exposure or CAR treatment in male (**J**) and female (**K**) rats. $I_h$-dependent voltage sag was attenuated by carrageenan, with a larger reduction occurring in male rats (**L**) than in female rats (**M**). Afterhyperpolarization (mAHP) was attenuated by AIE exposure, with a smaller reduction occurring in male rats (**N**) than in female rats (**O**). Data represent the mean ± SEM. Source data for all panels is included in **Figure 4—source data 1**. Data were analyzed using ANOVA,

*Figure 4 continued on next page*

Figure 4 continued

with exposure (AIR vs. AIE), treatment (CAR vs. SAL), sex, and injected current (when applicable) as factors. * indicates a significant difference between the related conditions; p < 0.05; $n$ = 10 rats/group.

The online version of this article includes the following source data for figure 4:

**Source data 1.** Numerical data for the intrinsic excitability and selected biophysical properties of PrL$^{PAG}$ neurons.

= 21.94, p < 0.0001, partial $\eta^2$ = 0.2335 [0.0817, 0.3827]). Notably, PrL$^{PAG}$ neurons from female rats displayed greater afterhyperpolarization than those from male rats in all treatment conditions except for AIE exposure paired with carrageenan (main effect of sex: $F_{(1,72)}$ = 12.92, p = 0.0006, partial $\eta^2$ = 0.1522 [0.0310, 0.2998]; AIE × sex × carrageenan interaction: $F_{(1,72)}$ = 6.81, p = 0.0110, partial $\eta^2$ = 0.0864 [0.0045, 0.2228]). The analysis revealed no further effects of treatment or sex on either hyperpolarization sag (main effect of AIE: $F_{(1,72)}$ = 0.03, p = 0.8528; AIE × carrageenan interaction: $F_{(1,72)}$ = 2.18, p = 0.1446; AIE × sex interaction: $F_{(1,72)}$ = 0.04, p = 0.8500; AIE × sex × carrageenan interaction: $F_{(1,72)}$ = 0.50, p = 0.4797) or afterhyperpolarization (main effect of carrageenan: $F_{(1,72)}$ = 0.04, p = 0.8333; AIE × carrageenan interaction: $F_{(1,72)}$ = 0.25, p = 0.6203; sex × carrageenan interaction: $F_{(1,72)}$ = 2.53, p = 0.1160).

## AIE exposure enhanced the E/I balance at inputs from the BLA onto PrL$^{PAG}$ neurons

The next set of studies assessed the impact of AIE and carrageenan on synaptic function at BLA inputs to PrL$^{PAG}$ neurons. This involved recording from green retrobead labeled pyramidal neurons in the PrL cortex while optically stimulating terminals from the BLA (*Figure 5A*).

To evaluate the E/I balance at BLA inputs to PrL$^{PAG}$ neurons, voltage-clamp recordings of optically evoked excitatory postsynaptic currents (oEPSCs) and optically evoked inhibitory postsynaptic currents (oIPSCs) were obtained from green retrobead labeled cells in the PrL. Analysis of oEPSC peak amplitudes indicated no significant effects of AIE, carrageenan, or sex (main effect of AIE: $F_{(1,69)}$ = 2.36, p = 0.1294; main effect of carrageenan: $F_{(1,69)}$ = 0.35, p = 0.5572; main effect of sex: $F_{(1,69)}$ = 3.64, p = 0.0604; AIE × carrageenan interaction: $F_{(1,69)}$ = 0.00, p = 0.9757; AIE × sex interaction: $F_{(1,69)}$ = 1.45, p = 0.2334; sex × carrageenan interaction: $F_{(1,69)}$ = 0.00, p = 0.9617; AIE × sex × carrageenan interaction: $F_{(1,69)}$ = 0.76, p = 0.3851; *Figure 5B, C*).

Conversely, oIPSC amplitude was significantly reduced in AIE exposed rats (main effect of AIE: $F_{(1,69)}$ = 37.66, p = 0.0000, partial $\eta^2$ = 0.3531 [0.1771, 0.4942]; *Figure 5D, E*). Additionally, carrageenan was found to enhance oIPSC amplitude (main effect of carrageenan: $F_{(1,69)}$ = 4.17, p = 0.0449, partial $\eta^2$ = 0.0570 [0.0000, 0.1858]); however, this increase was attenuated in AIE exposed rats (AIE × carrageenan interaction: $F_{(1,69)}$ = 5.16, p = 0.0262, partial $\eta^2$ = 0.0696 [0.0000, 0.2036]). No significant effects of sex, or interactions between sex and AIE or carrageenan on oIPSC amplitude were found (main effect of sex: $F_{(1,69)}$ = 0.83, p = 0.3644; AIE × sex interaction: $F_{(1,69)}$ = 2.90, p = 0.0929; sex × carrageenan interaction: $F_{(1,69)}$ = 0.97, p = 0.3287; AIE × sex × carrageenan interaction: $F_{(1,69)}$ = 1.84, p = 0.1788).

To quantify the resulting E/I balance at PrL$^{PAG}$ neurons, the ratios of oEPSCs to oIPSCs were compared. This revealed that AIE enhanced the E/I balance (main effect of AIE: $F_{(1,69)}$ = 52.48, p = 0.0000, partial $\eta^2$ = 0.4320 [0.2545, 0.5611]; *Figure 5F, G*) and female rats exhibited larger E/I ratios than male rats (main effect of sex: $F_{(1,69)}$ = 5.21, p = 0.0255, partial $\eta^2$ = 0.0703 [0.0000, 0.2045]). No significant effects of carrageenan or interactions between AIE, sex, and carrageenan were found (main effect of carrageenan: $F_{(1,69)}$ = 0.96, p = 0.3307; AIE × carrageenan interaction: $F_{(1,69)}$ = 0.01, p = 0.9158; AIE × sex interaction: $F_{(1,69)}$ = 0.05, p = 0.8190; sex × carrageenan interaction: $F_{(1,69)}$ = 1.72, p = 0.1935; AIE × sex × carrageenan interaction: $F_{(1,69)}$ = 0.08, p = 0.7741).

## The AMPA/NMDA ratio at direct inputs from the BLA onto PrL$^{PAG}$ neurons was unaltered by AIE exposure or carrageenan

To assess the AMPA/NMDA ratio at monosynaptic inputs from the BLA to PrL$^{PAG}$ neurons, voltage-clamp recordings of oAMPA and oNMDA currents were obtained from green retrobead labeled cells in the PrL during application of tetrodotoxin (TTX) and 4-aminopyridine (4-AP) (*Figure 6A*). Analysis of oAMPA current amplitude revealed no significant effects of AIE, carrageenan, or sex (main effect

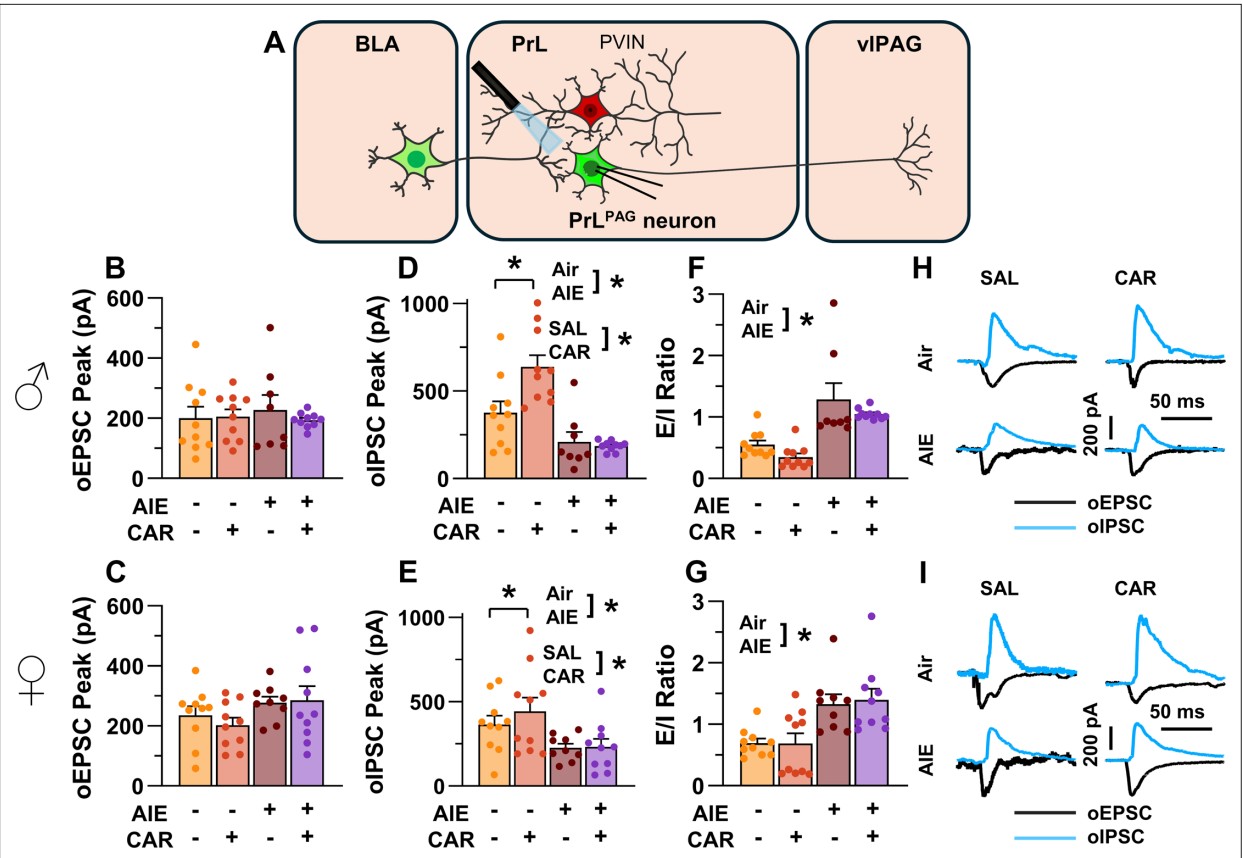

**Figure 5.** Optically evoked postsynaptic excitatory and inhibitory currents onto pyramidal neurons projecting from the prelimbic cortex to the ventrolateral periaqueductal gray (PrL$^{PAG}$). (**A**) Electrophysiological recordings were obtained from PrL$^{PAG}$ neurons in the PrL cortex. The amplitude of optically evoked excitatory postsynaptic currents (oEPSCs) was not significantly altered by adolescent intermittent ethanol (AIE) exposure or a carrageenan paw pain challenge (CAR) in either male (**B**) or female (**C**) rats. In contrast, the amplitude of optically evoked inhibitory postsynaptic currents (oIPSCs) was significantly reduced in both male (**D**) and female (**E**) AIE exposed rats. Carrageenan enhanced the amplitude of oIPSCs, but this increase was attenuated in AIE exposed rats. Examination of the oEPSC/oIPSC (excitation/inhibition, E/I) ratios as a measure of excitatory–inhibitory balance at basolateral amygdala (BLA) inputs onto PrL$^{PAG}$ neurons revealed that in AIE exposed animals, the E/I balance was significantly increased in both male (**F**) and female (**G**) rats. (**H**) Representative traces of the oEPSC and oIPSC currents recorded from male rats across all treatment groups. (**I**) Representative traces of oEPSC and oIPSC currents recorded from female rats across all treatment groups. Data represent the mean ± SEM. Source data for all panels is included in *Figure 5—source data 1*. Data were analyzed using ANOVA, with exposure (AIR vs. AIE), treatment (CAR vs. SAL), and sex as factors. * indicates a significant difference between the related conditions; p < 0.05; n = 8–10 rats/group.

The online version of this article includes the following source data for figure 5:

**Source data 1.** Numerical data characterizing optically evoked postsynaptic excitatory and inhibitory currents onto PrL$^{PAG}$ neurons.

of AIE: $F_{(1,72)} = 0.50$, p = 0.4817; main effect of carrageenan: $F_{(1,72)} = 0.31$, p = 0.5806; main effect of sex: $F_{(1,72)} = 3.19$, p = 0.0784; AIE × carrageenan interaction: $F_{(1,72)} = 0.03$, p = 0.8652; AIE × sex interaction: $F_{(1,72)} = 0.20$, p = 0.6576; sex × carrageenan interaction: $F_{(1,72)} = 0.29$, p = 0.5889; AIE × sex × carrageenan interaction: $F_{(1,72)} = 0.26$, p = 0.6112; *Figure 6B, C*). Likewise, evaluation of the amplitude of oNMDA currents revealed no effects of treatment condition or sex (main effect of AIE: $F_{(1,72)} = 1.25$, p = 0.2680; main effect of carrageenan: $F_{(1,72)} = 0.29$, p = 0.5932; main effect of sex: $F_{(1,72)} = 2.86$, p = 0.0952; AIE × carrageenan interaction: $F_{(1,72)} = 0.34$, p = 0.5635; AIE × sex interaction: $F_{(1,72)} = 0.42$, p = 0.5204; sex × carrageenan interaction: $F_{(1,72)} = 1.55$, p = 0.2176; AIE × sex × carrageenan interaction: $F_{(1,72)} = 2.40$, p = 0.1257; *Figure 6D, E*). The resultant ratios of oAMPA to oNMDA currents were compared and also were found to be unaltered by AIE, carrageenan, or sex (main effect of AIE: $F_{(1,72)} = 1.34$, p = 0.2513; main effect of carrageenan: $F_{(1,72)} = 1.00$, p = 0.3199; main effect of sex: $F_{(1,72)} = 2.81$, p = 0.0982; AIE × carrageenan interaction: $F_{(1,72)} = 0.25$, p = 0.6173; AIE × sex interaction: $F_{(1,72)} = 0.75$, p = 0.3907; sex × carrageenan interaction: $F_{(1,72)} = 0.01$, p = 0.9328; AIE × sex × carrageenan interaction: $F_{(1,72)} = 1.30$, p = 0.2582; *Figure 6F, G*).

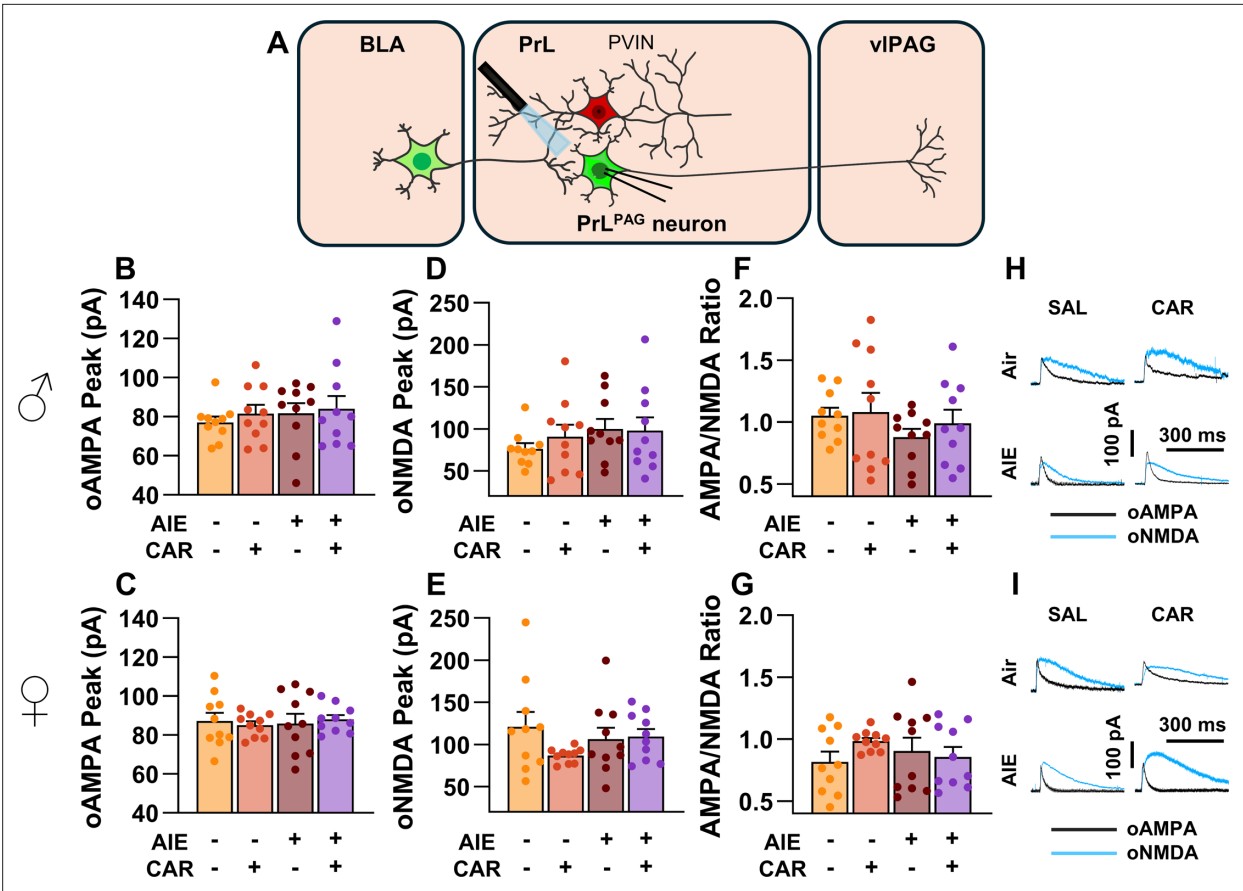

**Figure 6.** Optically evoked AMPA and NMDA currents at monosynaptic inputs from the basolateral amygdala (BLA) onto pyramidal neurons projecting from the prelimbic cortex to the ventrolateral periaqueductal gray (PrL^PAG). (**A**) Electrophysiological recordings were obtained from PrL^PAG neurons in the PrL cortex. The amplitude of optically evoked AMPA currents was not altered by adolescent intermittent ethanol (AIE) exposure or a carrageenan paw pain challenge (CAR) in either male (**B**) or female (**C**) rats. Similarly, the amplitude of optically evoked NMDA currents was unchanged across all treatment conditions in both male (**D**) and female (**E**) rats. The AMPA/NMDA ratio was also not significantly altered by AIE or CAR in male (**F**) or female (**G**) rats. (**H**) Representative traces of optically evoked AMPA and NMDA currents recorded from male rats across all treatment groups. (**I**) Representative traces of optically evoked AMPA and NMDA currents recorded from female rats across all treatment groups. Data represent the mean ± SEM. Source data for all panels is included in **Figure 6—source data 1**. Data were analyzed using ANOVA, with exposure (AIR vs. AIE), treatment (CAR vs. SAL), and sex as factors. $n$ = 10 rats/group.

The online version of this article includes the following source data for figure 6:

**Source data 1.** Numerical data characterizing optically evoked AMPA and NMDA currents at monosynaptic inputs from the basolateral amygdala (BLA) onto PrL^PAG neurons.

### Asynchronous excitatory postsynaptic currents at direct inputs from the BLA onto PrL^PAG neurons were unaffected by AIE exposure or carrageenan

To examine pre- and postsynaptic alterations in glutamatergic neurotransmission at BLA inputs onto PrL^PAG neurons, voltage-clamp recordings of optically evoked asynchronous excitatory postsynaptic currents (aEPSCs) were obtained from green retrobead labeled cells while bath applying TTX and 4-AP (**Figure 7A**). Analysis showed that female rats exhibited larger aEPSCs than male rats (main effect of sex: $F_{(1,70)}$ = 11.75, p = 0.0010, partial $\eta^2$ = 0.1437 [0.0256, 0.2925]). No other significant effects of AIE, carrageenan, or any interactions on aEPSC amplitude were observed (main effect of AIE: $F_{(1,70)}$ = 0.25, p = 0.6200; main effect of carrageenan: $F_{(1,70)}$ = 0.08, p = 0.7733; AIE × carrageenan interaction: $F_{(1,70)}$ = 0.24, p = 0.6239; AIE × sex interaction: $F_{(1,70)}$ = 0.09, p = 0.7713; sex × carrageenan interaction: $F_{(1,70)}$ = 3.62, p = 0.0613; AIE × sex × carrageenan interaction: $F_{(1,70)}$ = 0.47, p = 0.4974; **Figure 7B, C**). Likewise, assessment of the aEPSC interevent interval revealed no significant effects

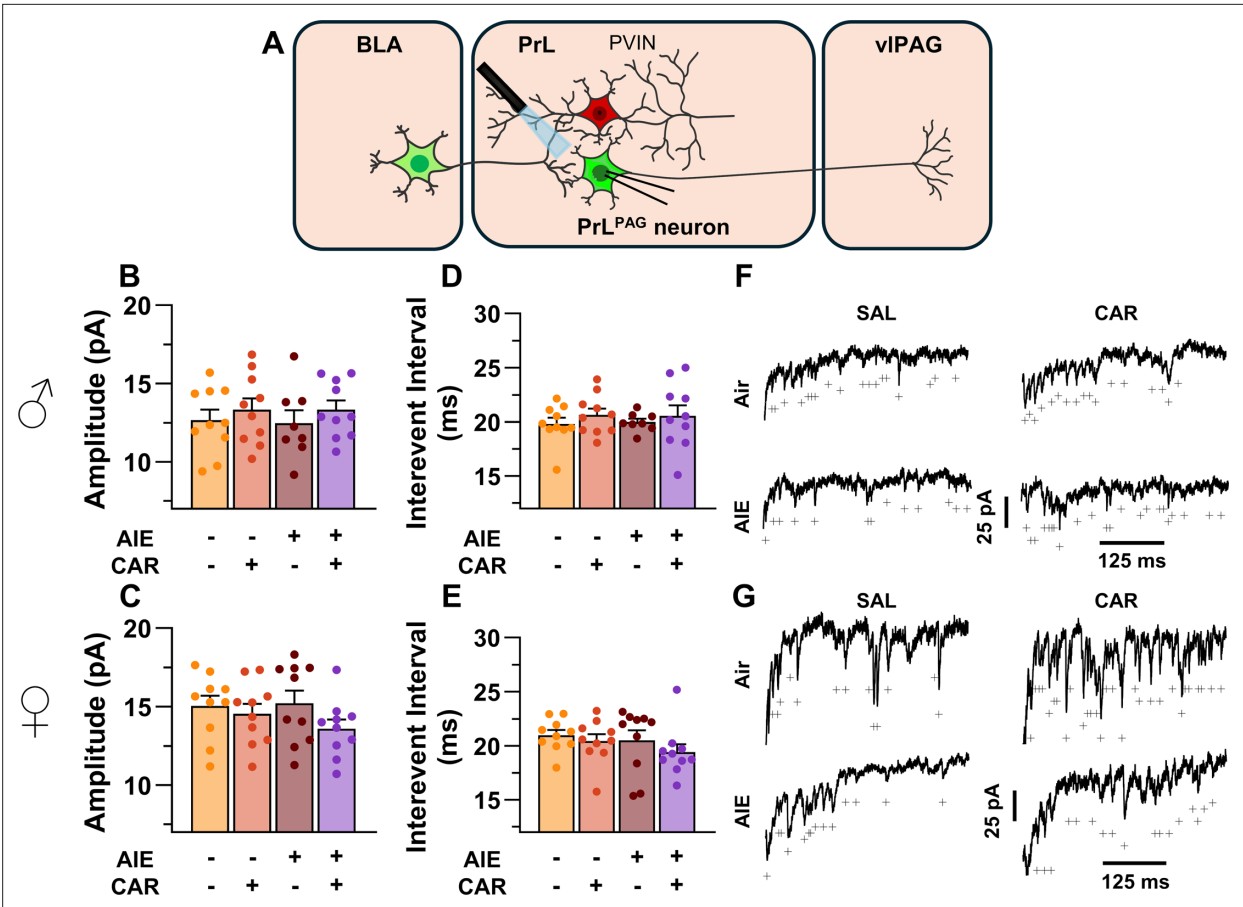

**Figure 7.** Optically evoked asynchronous excitatory postsynaptic currents (aEPSCs) at monosynaptic inputs from the basolateral amygdala (BLA) onto pyramidal neurons projecting from the prelimbic cortex to the ventrolateral periaqueductal gray (PrL^PAG). (**A**) Electrophysiological recordings were obtained from PrL^PAG neurons in the PrL cortex. When compared across treatment conditions, there were no differences in either the amplitude (**B, C**) or interevent interval (**D, E**) of aEPSCs. (**F**) Representative traces of aEPSCs recorded from male rats across all treatment groups. (**G**) Representative traces of aEPSCs recorded from female rats across all treatment groups. Data represent the mean ± SEM. Source data for all panels is included in *Figure 7—source data 1*. On the current traces, a + indicates an asynchronous event. Data were analyzed using ANOVA, with exposure (AIR vs. AIE), treatment (CAR vs. SAL), and sex as factors. $n$ = 8–10 rats/group.

The online version of this article includes the following source data for figure 7:

**Source data 1.** Numerical data characterizing optically evoked asynchronous excitatory postsynaptic currents (aEPSCs) at monosynaptic inputs from the basolateral amygdala (BLA) onto PrL^PAG neurons.

of AIE, carrageenan, or sex (main effect of AIE: $F_{(1,70)}$ = 0.52, p = 0.4731; main effect of carrageenan: $F_{(1,70)}$ = 0.01, p = 0.9106; main effect of sex: $F_{(1,70)}$ = 0.03, p = 0.8678; AIE × carrageenan interaction: $F_{(1,70)}$ = 0.15, p = 0.7021; AIE × sex interaction: $F_{(1,70)}$ = 0.65, p = 0.4224; sex × carrageenan interaction: $F_{(1,70)}$ = 2.39, p = 0.1265; AIE × sex × carrageenan interaction: $F_{(1,70)}$ = 0.02, p = 0.8815; *Figure 7D, E*).

## AIE exposure decreased and carrageenan increased the intrinsic excitability of PrL PVINs

After observing altered inhibition of PrL^PAG neurons, the impact of AIE exposure- and carrageenan-induced hyperalgesia on PrL PVIN intrinsic excitability was evaluated through current-clamp recordings of current evoked firing obtained from mCherry-tagged cells in the PrL cortex. As expected, mCherry-tagged neurons were distributed throughout layers II/III and V/VI in the PrL, with the highest concentration observed in layer V. *Table 2* contains a summary of the biophysical properties of the recorded PVINs. Analysis of these properties revealed a significant reduction in the AP threshold of PVINs from AIE exposed rats treated with carrageenan (AIE × carrageenan interaction: $F_{(1,72)}$ = 4.56, p = 0.0361, partial $\eta^2$ = 0.0596 [0.0000, 0.1865]) as well as greater afterhyperpolarization in PVINs from

**Table 2.** Biophysical properties of parvalbumin interneurons (PVINs) across treatment condition and sex.

| Condition | Sex | $V_{rest}$ (mV) | $R_{input}$ (MΩ) | $AP_{Thresh}$ (mV) | Sag ratio (%) | AHP (mV) |
|---|---|---|---|---|---|---|
| AIR: SAL | Male | −74.3 ± 1.9 | 145.1 ± 5.1 | −42.4 ± 0.7 | 2.6 ± 0.8 | 18.0 ± 0.9 |
| | Female | −72.6 ± 1.6 | 155.4 ± 8.8 | −42.3 ± 1.6 | 2.7 ± 0.4 | 15.3 ± 0.2 |
| AIR: CAR | Male | −72.5 ± 1.9 | 138.1 ± 7.4 | −42.0 ± 1.4 | 1.3 ± 0.5 | 19.4 ± 1.3 |
| | Female | −73.7 ± 1.6 | 135.4 ± 8.9 | −41.2 ± 0.8 | 2.3 ± 0.6 | 12.0 ± 1.2 |
| AIE: SAL | Male | −73.6 ± 1.3 | 155.7 ± 4.9 | −40.4 ± 0.6 | 1.2 ± 0.5 | 16.0 ± 1.8 |
| | Female | −74.6 ± 1.8 | 147.0 ± 5.2 | −41.4 ± 1.2 | 3.2 ± 0.8 | 14.1 ± 1.0 |
| AIE: CAR | Male | −71.6 ± 2.1 | 153.2 ± 8.3 | −44.3 ± 1.6* | 2.6 ± 0.7 | 16.1 ± 1.2 |
| | Female | −74.0 ± 1.6 | 141.3 ± 5.9 | −44.0 ± 2.0* | 2.5 ± 0.7 | 14.6 ± 1.8 |

* Denotes a significant difference; p < 0.05.

male rats (main effect of sex: $F_{(1,72)}$ = 14.51, p = 0.0003, partial $\eta^2$ = 0.1677 [0.0394, 0.3165]). No other significant differences between treatment conditions or sex were observed for these properties.

Analysis of the firing of PrL PVINs revealed AIE reduced intrinsic excitability, with the magnitude of this reduction increasing with the amount of current injected (main effect of AIE: $F_{(1,72)}$ = 5.41, p = 0.0228, partial $\eta^2$ = 0.0699 [0.0004, 0.2010]; AIE × current step interaction: $F_{(20,1440)}$ = 4.77, p = 0.0098, partial $\eta^2$ = 0.0621 [0.0289, 0.0744]; *Figure 8A–G*). Additionally, carrageenan enhanced the number of evoked APs at large but not small current steps, although the main effect was not significant (main effect of carrageenan: $F_{(1,72)}$ = 2.26, P = 0.1374; carrageenan × current step interaction: $F_{(20,1440)}$ = 3.57, p = 0.0304, partial $\eta^2$ = 0.0472 [0.0170, 0.0566]). There were no significant effects of sex, or interactions between treatment conditions, sex, and current step (main effect of sex: $F_{(1,72)}$ = 0.00, p = 0.9928; AIE × carrageenan interaction: $F_{(1,72)}$ = 0.08, p = 0.7720; AIE × sex interaction: $F_{(1,72)}$ = 0.18, p = 0.6685; sex × carrageenan interaction: $F_{(1,72)}$ = 0.01, p = 0.9110; AIE × sex × carrageenan interaction: $F_{(1,72)}$ = 0.00, p = 0.9700; sex × current step interaction: $F_{(20,1440)}$ = 0.43, p = 0.6532; AIE × sex × current step interaction: $F_{(20,1440)}$ = 0.80, p = 0.4510; AIE × carrageenan × current step interaction: $F_{(20,1440)}$ = 0.63, p = 0.5367; sex × carrageenan × current step interaction: $F_{(20,1440)}$=0.27, p = 0.7654; AIE × sex × carrageenan × current step interaction: $F_{(20,1440)}$ = 0.44, p = 0.6468). However, the number of current evoked APs increased with the amount of current injected (main effect of current step: $F_{(20,1440)}$ = 181.06, p = 0.000, partial $\eta^2$ = 0.7155 [0.6903, 0.7307]).

## AIE exposure decreased and carrageenan increased the E/I balance at inputs from the BLA onto PrL PVINs

After assessing the intrinsic excitability of PVINs, the next set of experiments characterized the impact of AIE exposure- and carrageenan-induced hyperalgesia on synaptic function at BLA inputs to PrL PVINs. This involved recording from mCherry-tagged PVINs in the PrL cortex while optically stimulating terminals from the BLA (*Figure 9A*).

To evaluate the E/I balance at BLA inputs to PVINs, voltage-clamp recordings of oEPSCs and oIPSCs were obtained from mCherry-tagged neurons in the PrL. Analysis revealed that AIE significantly reduced oEPSC amplitude (main effect of AIE: $F_{(1,72)}$ = 14.80, p = 0.0003, partial $\eta^2$ = 0.1705 [0.0409, 0.3194]; *Figure 9B, C*), with larger oEPSCs in females than males (main effect of sex: $F_{(1,72)}$ = 8.22, p = 0.0054, partial $\eta^2$ = 0.1025 [0.0094, 0.2429]). No significant effects of carrageenan, or interactions between AIE, carrageenan, or sex were observed (main effect of carrageenan: $F_{(1,72)}$ = 1.33, p = 0.2520; AIE × carrageenan interaction: $F_{(1,72)}$ = 0.08, p = 0.7839; AIE × sex interaction: $F_{(1,72)}$ = 0.26, p = 0.6126; sex × carrageenan interaction: $F_{(1,72)}$ = 0.41, p = 0.5254; AIE × sex × carrageenan interaction: $F_{(1,72)}$ = 0.26, p = 0.6110).

Examination of oIPSC amplitudes uncovered a significant AIE × sex interaction (AIE × sex interaction: $F_{(1,72)}$ = 4.48, p = 0.0377, partial $\eta^2$ = 0.0586 [0.0000, 0.1851]; *Figure 9D, E*), but no further significant effects of AIE, carrageenan, or sex (main effect of AIE: $F_{(1,72)}$ = 0.00, p = 0.9927; main effect of carrageenan: $F_{(1,72)}$ = 0.16, p = 0.6871; main effect of sex: $F_{(1,72)}$ = 0.14, p = 0.7094; AIE × carrageenan interaction: $F_{(1,72)}$ = 1.15, p = 0.2867; sex × carrageenan interaction: $F_{(1,72)}$ = 0.19, p =

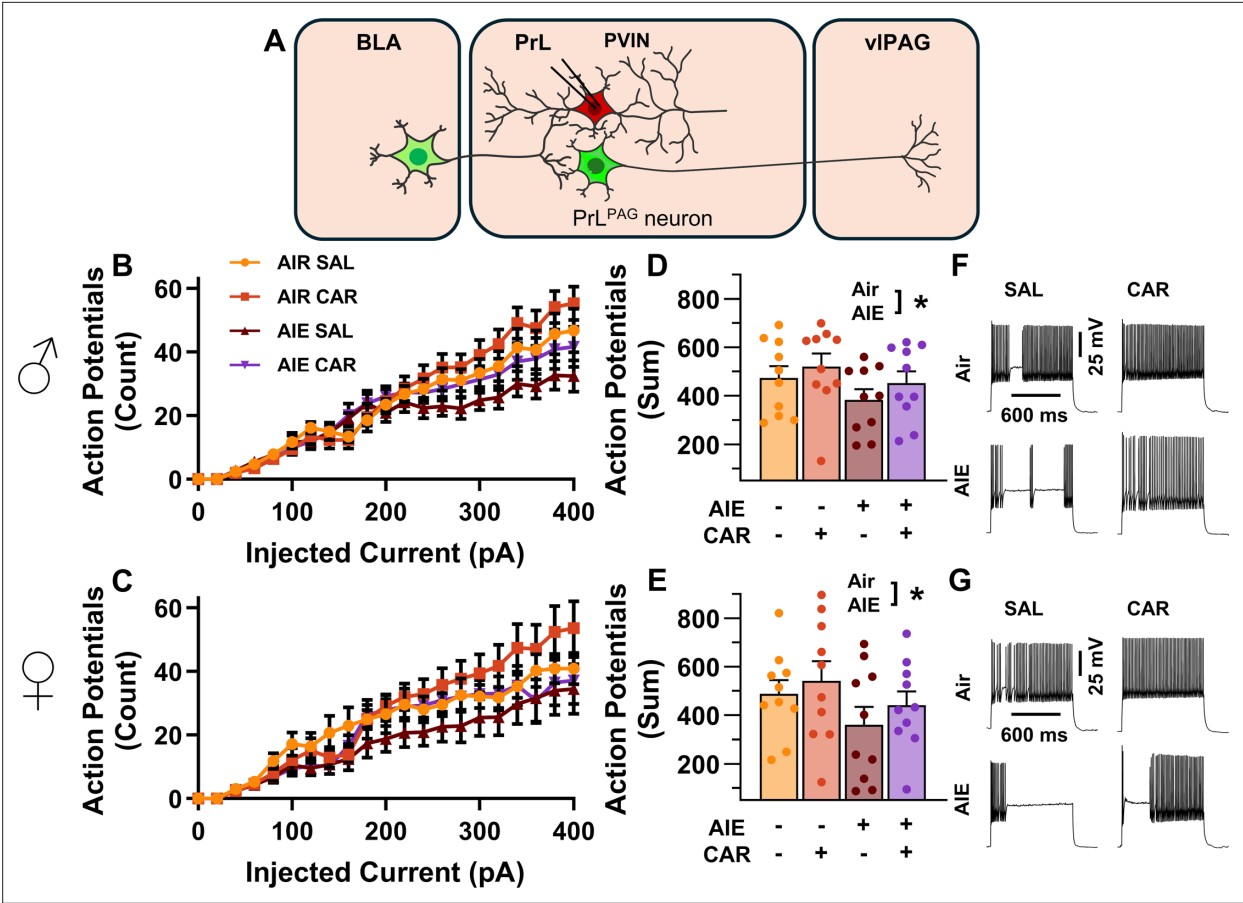

**Figure 8.** Intrinsic excitability of prelimbic (PrL) parvalbumin interneurons (PVINs). (**A**) Electrophysiological recordings were obtained from PVINs in the PrL cortex. (**B**) Depiction of the relationship between injected current and action potential firing in male rats across treatment conditions. (**C**) Depiction of the relationship between injected current and action potential firing in female rats across treatment conditions. (**D**) Adolescent intermittent ethanol (AIE) exposure reduced the cumulative number of action potentials fired across all current steps in male rats. (**E**) AIE exposure reduced the cumulative number of action potentials fired across all current steps in female rats. (**F**) Representative traces showing action potential spiking across treatment conditions in male rats. (**G**) Representative traces showing action potential spiking across treatment conditions in female rats. Data represent the mean ± SEM. Source data for all panels is included in *Figure 8—source data 1*. Data were analyzed using ANOVA, with exposure (AIR vs. AIE), treatment (CAR vs. SAL), sex, and injected current (when applicable) as factors. * indicates a significant difference between the related conditions; p < 0.05; n = 10 rats/group.

The online version of this article includes the following source data for figure 8:

**Source data 1.** Numerical data for the intrinsic excitability of prelimbic (PrL) parvalbumin interneurons (PVINs).

0.6673; AIE × sex × carrageenan interaction: $F_{(1,72)} = 0.07$, p = 0.7895). Subsequent post hoc analysis did not find significant effects of AIE in male (t = 1.49, p = 0.281) or female (t = −1.50, p = 0.274) rats nor did it find significant sex differences in Air (t = 1.23, p = 0.443) or AIE exposed (t = −1.76, p = 0.165) rats.

Comparisons of the oEPSC to oIPSC ratios indicated larger E/I ratios in females compared to males (main effect of sex: $F_{(1,72)} = 12.12$, p = 0.0009, partial $\eta^2 = 0.1440$ [0.0269, 0.2909]; *Figure 9F, G*). Additionally, AIE reduced the E/I ratio, with a more pronounced reduction observed in males than in females (main effect of AIE: $F_{(1,72)} = 15.60$, p = 0.0002, partial $\eta^2 = 0.1781$ [0.0453, 0.3273]; AIE × sex interaction: $F_{(1,72)} = 4.64$, p = 0.0345, partial $\eta^2 = 0.0606$ [0.0000, 0.1880]). In contrast, carrageenan augmented the E/I balance (main effect of carrageenan: $F_{(1,72)} = 7.48$, p = 0.0079, partial $\eta^2 = 0.0941$ [0.0067, 0.2326]). No further significant interactions between AIE, carrageenan, and sex were observed (AIE × carrageenan interaction: $F_{(1,72)} = 1.48$, p = 0.2278; sex × carrageenan interaction: $F_{(1,72)} = 0.57$, p = 0.4523; AIE × sex × carrageenan interaction: $F_{(1,72)} = 0.12$, p = 0.7347).

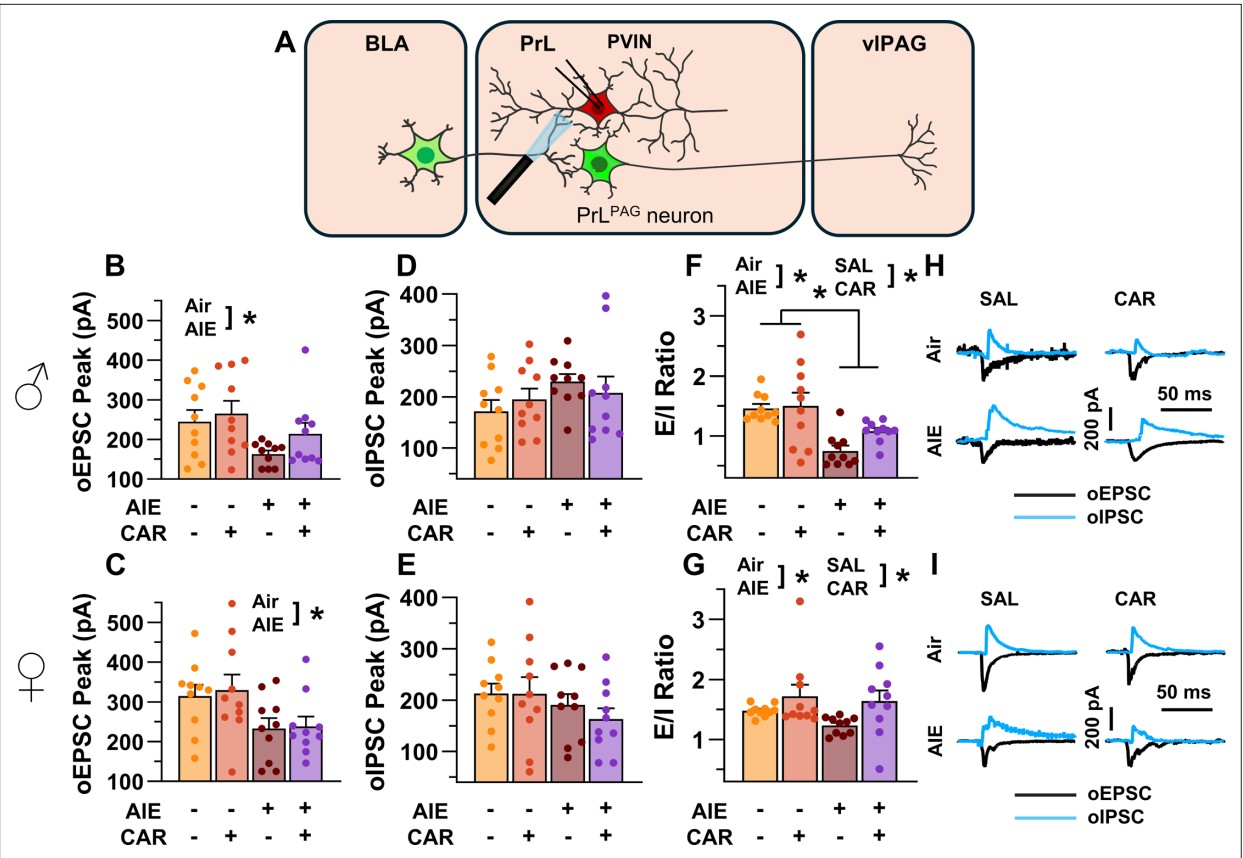

**Figure 9.** Optically evoked postsynaptic excitatory and inhibitory currents onto prelimbic (PrL) parvalbumin interneurons (PVINs). (**A**) Electrophysiological recordings were obtained from PVINs in the PrL cortex. The amplitude of optically evoked excitatory postsynaptic currents (oEPSCs) onto PVINs was found to be significantly reduced by adolescent intermittent ethanol (AIE) exposure in both (**B**) male and (**C**) female rats. Quantification of the amplitude of optically evoked inhibitory postsynaptic currents (oIPSCs) revealed that oIPSCs onto PVINs were altered by AIE in a sex-dependent manner, although post hoc analysis did not reveal a significant difference based on any combination of sex and AIE (**D, E**). Examination of the oEPSC/oIPSC (excitation/inhibition, E/I) ratios as a measure of excitatory–inhibitory balance at basolateral amygdala (BLA) inputs onto PVINs revealed that a carrageenan paw pain challenge (CAR) enhanced the E/I ratio at PVINs in both male (**F**) and female (**G**) rats, while AIE reduced the E/I ratio. The effect of AIE on E/I balance was greater in males (**F**) than in females (**G**). (**H**) Representative traces of the oEPSC and oIPSC currents recorded from male rats across all treatment groups. (**I**) Representative traces of oEPSC and oIPSC currents recorded from female rats across all treatment groups. Data represent the mean ± SEM. Source data for all panels is included in *Figure 9—source data 1*. Data were analyzed using ANOVA, with exposure (AIR vs. AIE), treatment (CAR vs. SAL), and sex as factors. * indicates a significant difference between the related conditions; p < 0.05; n = 10 rats/group.

The online version of this article includes the following source data for figure 9:

**Source data 1.** Numerical data characterizing optically evoked postsynaptic excitatory and inhibitory currents onto prelimbic (PrL) parvalbumin interneurons (PVINs).

## AIE exposure blunted carrageenan-induced increases in the AMPA/NMDA ratio at direct inputs from the BLA onto PrL PVINs

To assess the AMPA/NMDA ratio at monosynaptic BLA inputs onto PrL PVINs, voltage-clamp recordings of oAMPA and oNMDA currents were obtained from mCherry-tagged neurons in the PrL cortex during bath application of TTX and 4-AP (*Figure 10A*). Analysis revealed that AIE exposure attenuated oAMPA current amplitude, whereas carrageenan enhanced it (main effect of AIE: $F_{(1,72)} = 5.76$, p = 0.0190, partial $\eta^2 = 0.0740$ [0.0013, 0.2066]; main effect of carrageenan: $F_{(1,72)} = 4.44$, p = 0.0385, partial $\eta^2 = 0.0581$ [0.0000, 0.1844]; *Figure 10B, C*). No significant effects of sex or any interactions were observed (main effect of sex: $F_{(1,72)} = 0.28$, p = 0.5974; AIE × carrageenan interaction: $F_{(1,72)} = 1.32$, p = 0.2542; AIE × sex interaction: $F_{(1,72)} = 0.37$, p = 0.5458; sex × carrageenan interaction: $F_{(1,72)} = 0.15$, p = 0.7029; AIE × sex × carrageenan interaction: $F_{(1,72)} = 0.22$, p = 0.6428).

Similarly, the amplitude of oNMDA currents was reduced by AIE (main effect of AIE: $F_{(1,72)} = 7.32$, p = 0.0085, partial $\eta^2 = 0.0923$ [0.0062, 0.2303]; *Figure 10D, E*). However, neither carrageenan nor

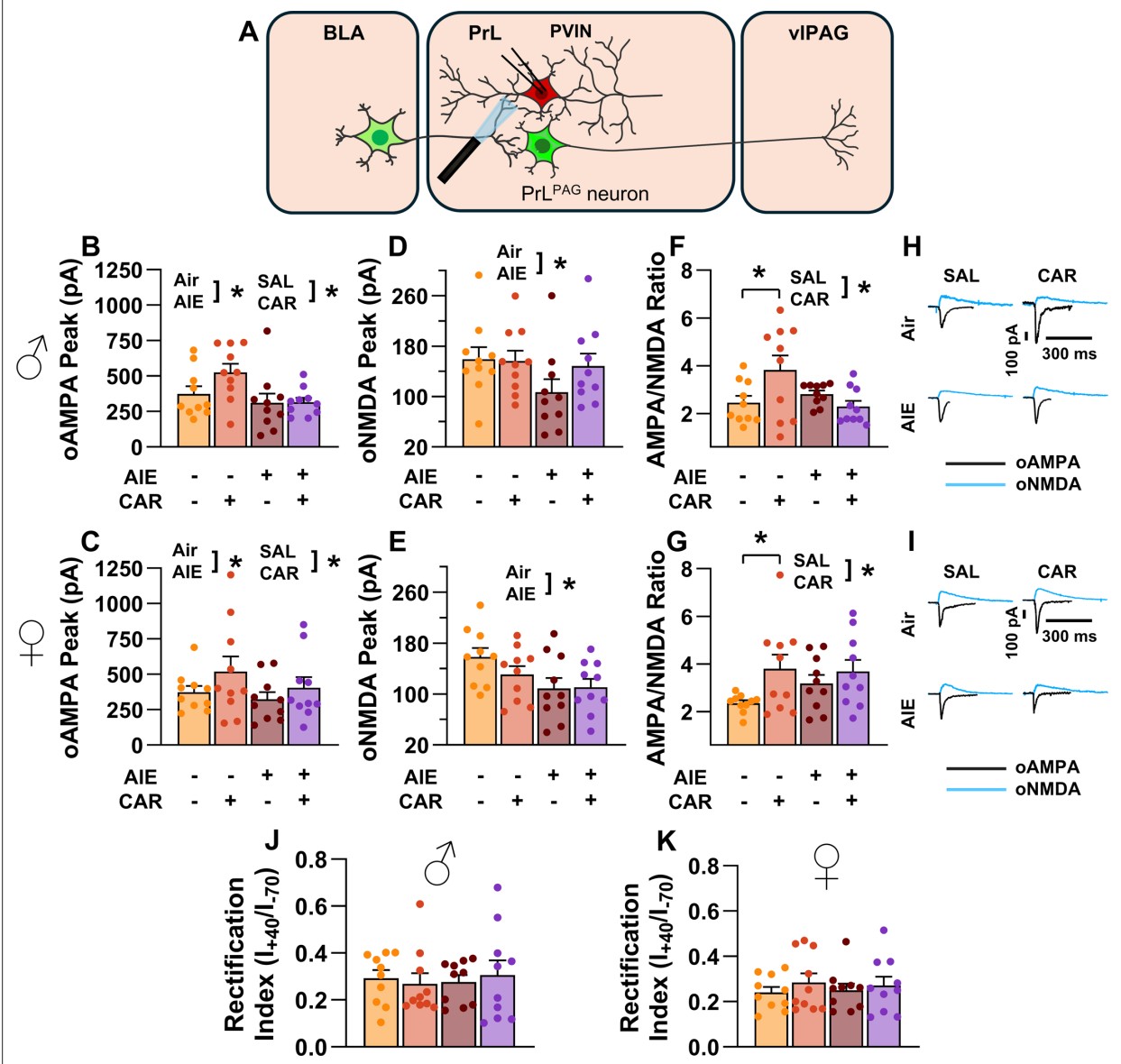

**Figure 10.** Optically evoked AMPA and NMDA currents at monosynaptic inputs from the basolateral amygdala (BLA) onto prelimbic (PrL) parvalbumin interneurons (PVINs). (**A**) Electrophysiological recordings were obtained from PVINs in the PrL cortex. The amplitude of optically evoked AMPA currents was increased by a carrageenan paw pain challenge (CAR) but decreased by adolescent intermittent ethanol (AIE) exposure in both male (**B**) and female (**C**) rats. Similarly, AIE reduced the amplitude of optically evoked NMDA currents in both male (**D**) and female (**E**) rats. Examination of the AMPA/NMDA ratios revealed that CAR enhanced the AMPA/NMDA ratio at BLA inputs onto PrL PVINs in both male (**F**) and female (**G**) rats. However, this increase was attenuated in AIE exposed rats. (**H**) Representative traces of optically evoked AMPA and NMDA currents recorded from male rats across all treatment groups. (**I**) Representative traces of optically evoked AMPA and NMDA currents recorded from female rats across all treatment groups. The rectification index was unchanged across treatment conditions in both male (**J**) and female (**K**) rats. Data represent the mean ± SEM. Source data for all panels is included in *Figure 10—source data 1*. Data were analyzed using ANOVA, with exposure (AIR vs. AIE), treatment (CAR vs. SAL), and sex as factors. * indicates a significant difference between the related conditions; $p < 0.05$; *n* = 10 rats/group.

The online version of this article includes the following source data for figure 10:

**Source data 1.** Numerical data characterizing optically evoked AMPA and NMDA currents at monosynaptic inputs from the basolateral amygdala (BLA) onto prelimbic (PrL) parvalbumin interneurons (PVINs).

sex, nor any interaction between AIE, carrageenan, or sex was observed to impact oNMDA currents (main effect of carrageenan: $F_{(1,72)}$ = 0.08, p = 0.7814; main effect of sex: $F_{(1,72)}$ = 1.69, p = 0.1971; interaction AIE × carrageenan: $F_{(1,72)}$ = 2.37, p = 0.1279; interaction AIE × sex: $F_{(1,72)}$ = 0.04, p = 0.8458; interaction sex × carrageenan: $F_{(1,72)}$ = 1.80, p = 0.1836; interaction AIE × sex × carrageenan: $F_{(1,72)}$ = 0.10, p = 0.7493).

After analyzing the oAMPA and oNMDA currents individually, ratios of oAMPA to oNMDA currents were compared. This revealed that carrageenan significantly enhanced the AMPA/NMDA ratio (main effect of carrageenan: $F_{(1,72)}$ = 6.28, p = 0.0145, partial $\eta^2$ = 0.0802 [0.0029, 0.2148]; *Figure 10F, G*). The analysis also uncovered a significant interaction between AIE and carrageenan, reflecting a reduction in the effect of carrageenan on AIE exposed rats (AIE × carrageenan interaction: $F_{(1,72)}$ = 6.47, p = 0.0131, partial $\eta^2$ = 0.0825 [0.0034, 0.2178]). No additional significant effects were detected (main effect of AIE: $F_{(1,72)}$ = 0.16, p = 0.6888; main effect of sex: $F_{(1,72)}$ = 2.21, p = 0.1411; AIE × sex interaction: $F_{(1,72)}$ = 2.90, p = 0.0928; sex × carrageenan interaction: $F_{(1,72)}$ = 1.00, p = 0.3203; AIE × sex × carrageenan interaction: $F_{(1,72)}$ = 0.73, p = 0.3953).

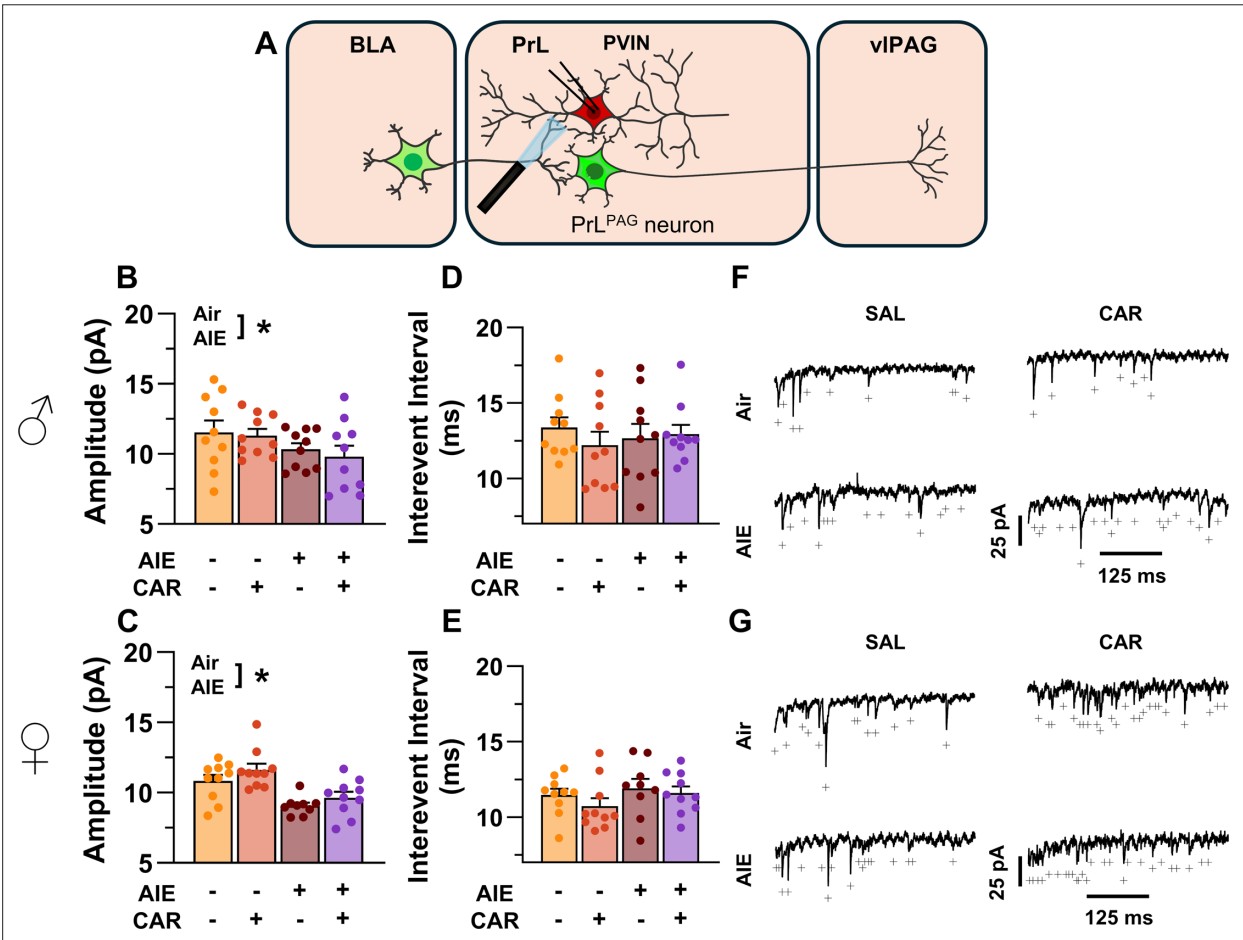

**Figure 11.** Optically evoked asynchronous excitatory postsynaptic currents (aEPSCs) at monosynaptic inputs from the basolateral amygdala (BLA) onto prelimbic (PrL) parvalbumin interneurons (PVINs). (**A**) Electrophysiological recordings were obtained from PVINs in the PrL cortex. Adolescent intermittent ethanol (AIE) exposure was found to decrease the amplitude of aEPSCs from both male (**B**) and female (**C**) rats. The interevent interval of aEPSCs, however, was unaltered by either AIE or a carrageenan paw pain challenge (CAR) in male (**D**) and female (**E**) rats. (**F**) Representative traces of aEPSCs recorded from male rats across all treatment groups. (**G**) Representative traces of aEPSCs recorded from female rats across all treatment groups. Data represent the mean ± SEM. + indicates an asynchronous event. Source data for all panels is included in *Figure 11—source data 1*. Data were analyzed using ANOVA, with exposure (AIR vs. AIE), treatment (CAR vs. SAL), and sex as factors. * indicates a significant difference between the related conditions; p < 0.05; *n* = 9–10 rats/group.

The online version of this article includes the following source data for figure 11:

**Source data 1.** Numerical data characterizing optically evoked asynchronous excitatory postsynaptic currents (aEPSCs) at monosynaptic inputs from the basolateral amygdala (BLA) onto prelimbic (PrL) parvalbumin interneurons (PVINs).

## AIE exposure reduced the amplitude of aEPSCs at direct inputs from the BLA onto PrL PVINs

To assess pre- and postsynaptic changes in glutamatergic neurotransmission at direct inputs from the BLA onto PVINs in the PrL cortex, voltage-clamp recordings were acquired from mCherry-tagged neurons during bath application of TTX and 4-AP (*Figure 11A*). Analysis revealed an AIE-induced reduction in amplitude (main effect of AIE: $F_{(1,71)}$ = 17.60, p = 0.0001, partial $\eta^2$ = 0.1986 [0.0572, 0.3493]; *Figure 11B, C*). No additional effects of carrageenan, sex, or interactions between treatment condition and sex were detected (main effect of carrageenan: $F_{(1,71)}$ = 0.15, p = 0.6961; main effect of sex: $F_{(1,71)}$ = 1.38, p = 0.2441; AIE × carrageenan interaction: $F_{(1,71)}$ = 0.10, p = 0.7556; AIE × sex interaction: $F_{(1,71)}$ = 0.44, p = 0.5115; sex × carrageenan interaction: $F_{(1,71)}$ = 1.89, p = 0.1733; AIE × sex × carrageenan interaction: $F_{(1,71)}$ = 0.00, p = 0.9505). In contrast, evaluation of the interevent interval revealed female rats had smaller interevent intervals than male rats (main effect of sex: $F_{(1,71)}$ = 8.44, p = 0.0049, partial $\eta^2$ = 0.1062 [0.0104, 0.2484]; *Figure 11D, E*), while finding no additional significant effects of AIE, carrageenan, or interactions between treatment conditions and sex (main effect of AIE: $F_{(1,71)}$ = 0.51, p = 0.4795; main effect of carrageenan: $F_{(1,71)}$ = 1.05, p = 0.3079; AIE × carrageenan interaction: $F_{(1,71)}$ = 1.00, p = 0.3196; AIE × sex interaction: $F_{(1,71)}$ = 0.46, p = 0.5009; sex × carrageenan interaction: $F_{(1,71)}$ = 0.00, p = 0.9530; AIE × sex × carrageenan interaction: $F_{(1,71)}$ = 0.29, p = 0.5943).

## Discussion

Emerging evidence from preclinical rodent models indicates that AIE exposure induces long-lasting hyperalgesia (*Bertagna et al., 2024*; *Kelley et al., 2024*; *Khan et al., 2023*; *Secci et al., 2024*). While changes within the extended amygdala circuitry contribute to altered nociception in AIE exposed animals (*Bertagna et al., 2024*; *Kelley et al., 2024*; *Secci et al., 2024*), the impact on prefrontal nociceptive circuits remains unexplored. This study investigated the effects of AIE exposure and carrageenan-induced inflammatory paw pain on synaptic function and intrinsic excitability within a BLA–PrL–vlPAG circuit involved in modulating the descending pain pathway. The central finding was that AIE enhanced mechanical allodynia, and that this enhancement was accompanied by altered E/I balance and intrinsic excitability at PrL^PAG neurons and PVINs (*Figure 12*). Carrageenan-induced hyperalgesia was unaltered by AIE exposure while pain-induced plasticity at BLA inputs to PrL PVINs was reduced.

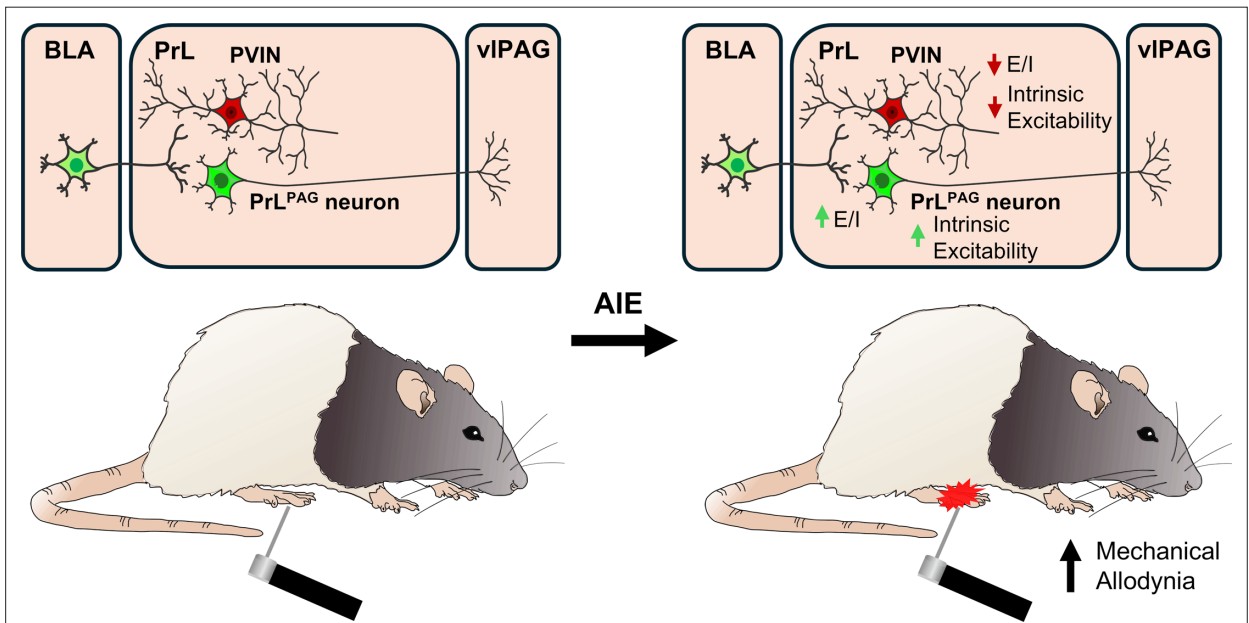

**Figure 12.** Visual summary of the principal findings regarding the effects of adolescent intermittent ethanol (AIE) exposure on basolateral amygdala–prelimbic–ventrolateral periaqueductal gray (BLA–PrL–vlPAG) circuitry and mechanical allodynia.

In this study, mechanical and thermal sensitivity were assessed from adolescence to early adulthood using the electronic Von Frey and Hargreaves tests. Mechanical sensitivity was heightened in both male and female rats exposed to AIE, consistent with previous findings in rodents (*Bertagna et al., 2024*; *Kelley et al., 2024*; *Khan et al., 2023*; *Secci et al., 2024*). While AIE-induced significant mechanical hypersensitivity, thermal sensitivity remained unchanged. Although this contrasts with previous research (*Khan et al., 2023*; *Secci et al., 2024*), for female rats this difference may be due, at least in part, to different statistical approaches for analysis of the data. In the present study, sex was included as a between-subjects factor in the ANOVA, whereas the prior study analyzed for male and female rat data separately. Consistent with this, when we analyzed our male and female data separately, we did observe a transient increase in thermal sensitivity in female rats as was reported previously (*Secci et al., 2024*). Discrepancies regarding the impact of AIE on thermal sensitivity in males may stem from methodological differences such as the use of different strains of rats and differences in the ethanol exposure paradigms.

After assessing the effects of AIE exposure on mechanical sensitivity, patch-clamp slice electrophysiology was performed to assess intrinsic excitability and synaptic function within the BLA–PrL–vlPAG circuit. For PrL$^{PAG}$ neurons, these experiments revealed that AIE increased intrinsic excitability. Previous research has shown both unchanged (*Galaj et al., 2020*; *Obray et al., 2022*; *Trantham-Davidson et al., 2017*) and increased intrinsic excitability of PrL pyramidal neurons following AIE (*Galaj et al., 2020*) or voluntary alcohol consumption during adolescence (*Salling et al., 2018*). As pyramidal neurons in the PrL cortex can be classified as intratelencephalic (IT) or extratelencephalic (ET) based on their projection targets (*Anastasiades and Carter, 2021*; *Baker et al., 2018a*) and as each of these populations display unique physiological properties (*Baker et al., 2018b*; *Dembrow et al., 2010*), one potential explanation for these results is that adolescent ethanol exposure differentially effects these populations. Consistent with this hypothesis, IT pyramidal neurons projecting from the PrL to the BLA and the nucleus accumbens do not display altered intrinsic excitability following AIE exposure (*Obray et al., 2022*) whereas PT PrL$^{PAG}$ neurons do. Notably, PT neurons display greater $I_h$-dependent voltage sag than IT neurons, and adolescent alcohol exposure has been reported to reduce $I_h$ current (*Salling et al., 2018*), suggesting a possible mechanism for greater ethanol effects on intrinsic excitability in ET neurons than in IT neurons. However, for PrL$^{PAG}$ neurons in the present study there was not a significant AIE-induced reduction in voltage sag. This disparity could be related to several methodological differences between studies, including differences in the exposure method (passive vapor versus voluntary drinking), the age of the animals at testing, and the use of rats versus mice. Regardless, it remains possible AIE may differentially affect the intrinsic excitability of ET and IT pyramidal neurons in the PrL.

Increased intrinsic excitability of PrL$^{PAG}$ neurons was accompanied by augmented E/I balance in AIE exposed rats. The increased E/I balance resulted from reduced oIPSC amplitude. This indicated that AIE exposure reduced BLA-driven feedforward inhibition of PrL$^{PAG}$ neurons. This is an interesting observation as it suggests that AIE prevents the normal developmental shift toward greater inhibition at PrL pyramidal neurons that occurs as PVINs mature during adolescence (*Caballero et al., 2014*; *Caballero et al., 2021*; *Du et al., 2018*; *Klune et al., 2021*).

To investigate the change in feedforward inhibition, we conducted electrophysiological recordings from PrL PVINs. These recordings revealed significantly reduced intrinsic excitability in AIE exposed animals, consistent with previous reports (*Trantham-Davidson et al., 2017*). Notably, PVIN intrinsic excitability is developmentally regulated and increases during adolescence (*Koppensteiner et al., 2019*). Decreased intrinsic excitability coincided with reduced BLA-driven E/I balance at PrL PVINs in male AIE exposed rats, with an attenuated effect in females. The reduced E/I balance primarily stemmed from a decrease in oEPSC amplitude, although AIE did alter oIPSC amplitude in a sex-dependent manner. The sex-dependent effect of AIE on oIPSCs is difficult to interpret, as post hoc tests did not reveal any significant effects. However, visual inspection of the data suggests a potential trend toward increased oIPSC amplitude in male AIE exposed rats and reduced amplitude in females. Intriguingly, voluntary alcohol consumption during adolescence elicits sex-dependent effects on PrL somatostatin interneuron intrinsic excitability (*Sicher et al., 2023*). As these neurons receive inputs from the BLA (*Cummings and Clem, 2020*; *McGarry and Carter, 2016*) and project to PrL PVINs (*Cummings and Clem, 2020*), it is possible that somatostatin neurons are responsible for the sex-dependent effects of AIE on E/I balance and oIPSCs at PVINs.

To better characterize the reduction in glutamate signaling, we next measured monosynaptic oAMPA and oNMDA currents at BLA inputs onto PVINs. This revealed that AIE significantly decreased oAMPA and oNMDA currents without altering the overall AMPA/NMDA ratio. This agreed with prior research from our lab which found AIE reduced electrically evoked AMPA and NMDA currents onto PVINs (*Trantham-Davidson et al., 2017*). To determine whether this reduction resulted from pre- or postsynaptic changes, aEPSCs were recorded from PVINs in the PrL cortex. The amplitude but not the interevent interval of the aEPSCs was reduced in AIE exposed animals, indicating a reduction in postsynaptic receptor function. These findings provide strong evidence for reduced glutamate signaling efficacy at BLA inputs onto PVINs following AIE exposure. It was also revealed that following AIE exposure, BLA-dependent feedforward inhibition of PrL$^{PAG}$ neurons was decreased. This reduction may have resulted, at least in part, from the observed decreases in PrL PVIN intrinsic excitability and postsynaptic glutamate receptor function at BLA inputs to PrL PVINs. Within the BLA–PrL–vlPAG circuit, PrL$^{PAG}$ neuron activation is generally associated with antinociceptive effects (*Drake et al., 2021*; *Gadotti et al., 2019*; *Gao et al., 2023*; *Huang et al., 2019*; *Yin et al., 2020*), while PrL PVIN activation is associated with pronociceptive effects (*Zhang et al., 2015*). However, PrL$^{PAG}$ neuron activation can be pronociceptive (*Fan et al., 2018*). Notably, chronic activation of a PrL nociceptive ensemble including PrL$^{PAG}$ neurons has been shown to induce chronic pain-like behaviors (*Qi et al., 2022*). As PrL$^{PAG}$ neurons project to both GABA and glutamate neurons in the vlPAG (*Guo et al., 2023*; *Huang et al., 2019*), and these populations have been shown to be pro- and antinociceptive, respectively (*Samineni et al., 2017*), the balance of PrL$^{PAG}$ input onto these populations may determine the effect of PrL$^{PAG}$ activation on nociception. While speculative, we hypothesize that AIE selectively strengthens the input from PrL$^{PAG}$ neurons onto vlPAG GABA neurons, resulting in mechanical allodynia that could be alleviated by inhibiting PrL$^{PAG}$ neurons or activating PrL PVINs. Alternatively, these changes could represent an antinociceptive compensatory response for pronociceptive changes in other nociceptive circuits. Future experiments could test these hypotheses.

The intrinsic excitability of PrL$^{PAG}$ neurons was increased following a carrageenan-induced pain challenge, akin to the heightened excitability observed in AIE exposed rats. Intrinsic excitability is also enhanced in PrL pyramidal neurons 3–5 days after intraplantar injection with complete Freund's adjuvant (*Wu et al., 2016*). In addition to enhancing the excitability of PrL$^{PAG}$ neurons, we observed that carrageenan also increased the amplitude of oIPSCs in Air control rats, with a blunted effect in AIE exposed rats. Despite no change in glutamate transmission at BLA inputs onto PrL$^{PAG}$ neurons, the E/I balance remained unaltered. These findings are similar to those seen in a kaolin-carrageenan arthritis model, where the amplitude of BLA-driven electrically evoked IPSCs (eIPSCs) onto PrL pyramidal neurons is significantly increased without a corresponding change in glutamate transmission (*Ji et al., 2010*). These findings indicate that enhanced inhibitory transmission at PrL pyramidal neurons is a hallmark of carrageenan-induced inflammatory pain.

As with PrL$^{PAG}$ neurons, carrageenan elevated the intrinsic excitability of PVINs, albeit only at large current steps. Accompanying this elevation was enhanced E/I balance at PVINs. There was not a significant change in either oEPSC or oIPSC amplitude, suggesting subtle alterations in both excitatory and inhibitory neurotransmission. The monosynaptic oAMPA current was enhanced at PrL PVINs, leading to an increase in the AMPA/NMDA ratio of Air control rats, with a weakened effect observed in AIE exposed rats. Surprisingly, the increased oAMPA currents at PVINs did not correspond to changes in either aEPSC amplitude or interevent interval. This leaves it unclear whether the observed change in synaptic strength resulted from a modification in presynaptic release probability or postsynaptic receptor function.

The present study demonstrates that in carrageenan-induced inflammatory pain, BLA inputs onto PVINs are strengthened. This strengthening, alongside increased PVIN excitability, enhances inhibition of PrL$^{PAG}$ neurons. Notably, PrL$^{PAG}$ neuron intrinsic excitability is increased, possibly to compensate for increased inhibitory input. Remarkably, the effects of carrageenan on synaptic function were blunted in AIE exposed rats, suggesting that AIE not only induces PVIN hypofunction but also restricts pain-induced plasticity at PVINs. Intriguingly, AIE augments the number of perineuronal net (PNN) enwrapped PVINs in the PrL cortex (*Dannenhoffer et al., 2022*; *Obray et al., 2025*). As PNNs play a role in stabilizing synapses and restricting plasticity (*Cornez et al., 2020*; *Pizzorusso et al., 2002*), this may reduce synaptic plasticity at PrL PVINs. Surprisingly, despite restricted plasticity at BLA inputs to PVINs, there were no AIE-dependent changes in carrageenan-induced hyperalgesia.

In conclusion, the present study examined the effects of AIE exposure and a carrageenan pain challenge on a BLA–PrL–vlPAG circuit involved in modulating the descending pain pathway. Following AIE exposure, while rats displayed enhanced mechanical sensitivity, carrageenan-induced hyperalgesia was not altered by a history of AIE exposure. In AIE exposed rats, BLA inputs to the PrL were biased toward decreased feedforward inhibition of PrL$^{PAG}$ neurons. A carrageenan pain challenge increased BLA-mediated inhibitory drive onto these neurons in Air control but not AIE exposed animals. These changes suggest that AIE induces long-lasting reductions in feedforward inhibition of PrL$^{PAG}$ neurons which accompany increased mechanical sensitivity. In addition to reduced feedforward inhibition, AIE may also diminish plasticity at PrL PVINs. Beyond the implications for nociception, these AIE-induced alterations in BLA–PrL–vlPAG function may impact other PrL$^{PAG}$ neuron-mediated behaviors such as context fear discrimination (*Rozeske et al., 2018*), passive avoidance (*Johnson et al., 2022*), and arousal (*Guo et al., 2023*).

# Materials and methods

## Key resources table

| Reagent type (species) or resource | Designation | Source or reference | Identifiers | Additional information |
|---|---|---|---|---|
| Strain, strain background (*Rattus norvegicus*) | Parvalbumin-Cre on a Long-Evans background | Rat Resource and Research Center | RRID:RRRC_00773 | Male and female |
| Recombinant DNA reagent | AAV5-hSyn-hChR2(H134R)-EYFP | Addgene | RRID:Addgene_26973 Cat#:26973-AAV5 | |
| Recombinant DNA reagent | AAV2-hSyn-DIO-mCherry | Addgene | RRID:Addgene_50459 Cat#:50459-AAV2 | |
| Chemical compound, drug | $\lambda$-Carrageenan (low viscosity) | Tokyo Chemical Industry | Cat#:C2871 CAS#:9064-57-7 | |
| Chemical compound, drug | Picrotoxin | Ascent Scientific | Cat#:ASC-315 CAS#:124-87-8 | |
| Chemical compound, drug | Kynurenic acid sodium salt | Hello Bio | Cat#:HB0363 CAS#:2439-02-3 | |
| Chemical compound, drug | Tetrodotoxin (citrate) | Cayman Chemical | Cat#:14964 CAS#:18660-81-6 | |
| Chemical compound, drug | 4-Aminopyridine | Sigma-Aldrich | Cat#:A78403 CAS#:504-24-5 | |
| Chemical compound, drug | DL-APV | Cayman Chemical | Cat#:14540 CAS#:76326-31-3 | |
| Software, algorithm | Axograph X | Axograph | RRID:SCR_014284 | |
| Software, algorithm | Stata 15.1 | StataCorp | RRID:SCR_012763 | |
| Other | Green Retrobeads IX | Lumafluor | | |

## Animals

Parvalbumin-Cre rats on a Long-Evans background were obtained from the Rat Resource and Research Center (line #00773) and bred to establish a colony at the Medical University of South Carolina. Rats were genotyped at postnatal day (PD) 14, with only hemizygote rats included in this study. On PD 21, rat pups were weaned, same sex group-housed (2–3 per cage) and assigned to one of four treatment groups: Air-saline ($n$ = 30), Air-carrageenan ($n$ = 30), AIE-saline ($n$ = 30), or AIE-carrageenan ($n$ = 29). Rats were housed in a temperature and humidity-controlled environment on a 12/12-hr light/dark cycle, with lights off from 09:00 to 21:00 each day. Teklad 2918 (Envigo, Indianapolis, IN) chow and water were provided to the rats ad libitum. All procedures were carried out in accordance with the National Research Council's Guide for the Care and Use of Laboratory Animals (2011) and were approved by the Medical University of South Carolina Institutional Animal Care and Use Committee.

## AIE exposure

AIE exposure was carried out as previously described (*Chandler et al., 2022*; *Gass et al., 2014*). All rats underwent eight cycles of intermittent ethanol vapor exposure beginning at PD 28 and continuing

through PD 54. Each cycle consisted of 2 days of ethanol vapor exposure separated by 2 days of no ethanol exposure. The litter-matched Air rats received the same treatment except they were not exposed to ethanol vapor. On ethanol exposure days, rats were placed in the chambers at 18:00 and removed from the chambers at 08:00 on the following day. Upon removal from the chambers, the level of intoxication of each rat was rated on the following 5-point behavioral intoxication scale: 1 = no signs of intoxication; 2 = slightly intoxicated (slight motor impairment); 3 = moderately intoxicated (obvious motor impairment but able to walk); 4 = highly intoxicated (dragging abdomen, loss of righting reflex); 5 = extremely intoxicated (loss of righting reflex, and loss of eye blink reflex) (*Barker et al., 2017*; *Gass et al., 2014*; *Glover et al., 2021*). On the last day of each cycle, blood was collected from the tail vein and analyzed for BEC using an Analox alcohol analyzer (AM1, Analox Instruments, Stourbridge, GBR). Following the last exposure cycle, the rats remained group housed in the vivarium until undergoing surgery at ~PD 80, after which they were single housed until being sacrificed to obtain slices for experimental use.

## Stereotaxic surgery

Rats undergoing stereotaxic surgery were induced and maintained under a surgical plane of anesthesia using isoflurane (2–3%). Intracranial injections were performed on a Kopf rat stereotaxic instrument (Kopf Instruments, Tujunga, CA, USA). A Micro4 (World Precision Instruments [WPI], Sarasota, FL, USA) controlled UMP3 microinjection pump (WPI) connected to a glass syringe (80100, Hamilton Company, Reno, NV) under stereotaxic control was used to inject 500 nl green retrobeads IX (Lumafluor Inc, Durham, NC) into the left vlPAG (from bregma: –8.4 mm AP, –0.7 mm ML, –6.3 mm DV), 500 nl of AAV5-hSyn-hChR2(H134R)-EYFP (26973-AAV5, Addgene, Watertown, MA) into the left BLA (from bregma: –2.7 mm AP, –5.1 mm ML, –8.8 mm DV), and 750 nl of AAV2-hSyn-DIO-mCherry (50459-AAV2, Addgene) into the left PrL cortex (from bregma: +2.8 mm AP, –0.6 mm ML, –3.8 mm DV). All injections occurred at 1 nl/s with the injector remaining in place for an additional 5 min following completion of the infusion. After surgery, a minimum of 4 weeks was given to allow the rats to recover and for retrograde transport and viral expression to occur.

## Assessment of mechanical and thermal sensitivity

Mechanical and thermal sensitivity were assessed using the electronic Von Frey and Hargreaves tests, respectively. During the 3 days leading up to the first assessment rats were habituated to handling. On the day before the first assessment rats were acclimated to the testing apparatuses. This acclimation session consisted of 15-min exposure to the electronic Von Frey enclosure followed by 30-min exposure to the Hargreaves enclosure. Assessments began on PD 24 prior to initiation of ethanol vapor exposure on PD 28 and continued every 7 days until PD 80. Rats that subsequently underwent stereotaxic surgery followed by 4 weeks of recovery were reassessed for pain sensitivity. This was followed by injection with 100 µl of carrageenan (C2871, Tokyo Chemical Industry, Tokyo, JPN; 1% wt/vol in saline) or saline in the right hindpaw under brief isoflurane anesthesia. Subsequent pain sensitivity tests were conducted at 2-, 6-, and 24-hr post-injection, after which rats were sacrificed to obtain slices for experimental use.

### Electronic Von Frey test for mechanical allodynia

On pain sensitivity test days, rats were placed in a 17″ × 8.5″ × 10″ Plexiglas enclosure with a metal mesh floor. After a 5-min acclimation period, the mechanical sensitivity of the right hindpaw was assessed using an electronic Von Frey unit (38450, Ugo Basile, Gemonio, ITA). This consisted of placing the electronic Von Frey filament perpendicular to the plantar surface of the hindpaw and applying force until a sharp withdrawal response was elicited. Mechanical sensitivity was assessed three times per session for each rat and the average gram-force required to elicit a withdrawal response was recorded.

### Hargreaves test for thermal hyperalgesia

Following assessment of allodynia, rats were placed in a 4″ × 8″ × 5.5″ plexiglass enclosure with a glass floor. After a 15-min habituation period, the right hindpaw was stimulated using an infrared emitter from Ugo Basile (37570; 60% maximum intensity) and the latency to hindpaw withdrawal was measured. A 30-s cutoff was used to prevent tissue damage in rats that were unresponsive to the

thermal stimulus. In each session, the paw withdrawal latency was measured three times per rat and the average score was used to quantify the level of thermal sensitivity.

## Electrophysiological recordings

Acute slices were obtained from rats for electrophysiological recordings beginning at PD 110. Current-clamp experiments were performed as previously described (*Trantham-Davidson et al., 2014*; *Trantham-Davidson et al., 2017*). In brief, rats were anesthetized with isoflurane, and the brain was rapidly removed and placed into ice-cold cutting solution containing (in mM): 93 NMDG, 2.5 KCl, 1.2 $NaH_2PO_4$, 30 $NaHCO_3$, 10 D-glucose, 20 HEPES, 2.5 $C_5H_9NO_3S$, 5 ascorbic acid, 15 sucrose, 10 $MgCl_2$, and 0.5 $CaCl_2$. Following sectioning using a Leica vibratome (VT 1200S, Wetzlar, DEU), 280 µM thick slices were incubated for at least 60 min at 34°C in artificial cerebrospinal fluid (aCSF) containing (in mM): 92 NaCl, 2.5 KCl, 1.2 $NaH_2PO_4$, 30 $NaHCO_3$, 10 D-glucose, 20 HEPES, 5 ascorbic acid, 10 $MgCl_2$, and 0.5 $CaCl_2$. After incubation, slices were transferred to a submerged recording chamber held at 34°C and constantly perfused with recording aCSF containing (in mM): 125 NaCl, 2.5 KCl, 25 $NaHCO_3$, 10 D-glucose, 0.4 ascorbic acid, 1.3 $MgCl_2$, and 2 $CaCl_2$. Each of these solutions was pH adjusted (pH 7.3–7.43), with an osmolarity of 300–310 mOsm, and was continuously aerated with 95% $O_2$/5% $CO_2$.

Recordings were performed using a Multiclamp 700B amplifier (Molecular Devices, San Jose, CA) connected to a Windows-PC running Axograph X software through an ITC-18 digital to analog converter (HEKA Instruments, Holliston, MA). A Sutter Instruments P-1000 micropipette puller (Novato, CA) was used to pull borosilicate glass electrodes. Tip resistances ranged from 4 to 8 MΩ. All recordings were obtained from visually identified green retrobead labeled PrL[PAG] neurons or mCherry-tagged PVINs in the left PrL. Cells were identified using a Zeiss Axio Examiner.A1 microscope (Oberkochen, DEU) equipped with a DIC filter and a filter for visualizing green retrobeads and mCherry. All internal solutions were adjusted to pH 7.4 and 285 mOsm. Series and membrane resistance were measured at the beginning and end of each recording, and if the series resistance exceeded 20 MΩ or changed by more than 10% then the cell was excluded from the analysis.

### Current-clamp recordings

Electrodes were filled with an internal solution containing (in mM): 125 potassium gluconate, 20 KCl, 10 HEPES, 1 EGTA, 2 $MgCl_2$, 2 $Na_2$-ATP, 0.3 Tris-GTP, and 10 phosphocreatine. To assess intrinsic excitability, picrotoxin (100 µM) and kynurenic acid (2 mM) were added to the perfused aCSF and 1-s current steps were applied in 20 pA increments ranging from –100 to +400 pA. Recordings were digitized at 10 kHz, filtered at 2 kHz, and analyzed for the number of APs elicited by each current step.

### Voltage-clamp recordings

Electrodes were filled with an internal solution containing (in mM): 125 cesium methanesulfonate, 10 CsCl, 4 NaCl, 10 HEPES, 1 EGTA, 2 $MgCl_2$, 2 $Na_2$-ATP, 0.5 Tris-GTP, 10 phosphocreatine, and 1 QX-314-Cl. All voltage-clamp recordings were digitized at 10 kHz and filtered at 2 kHz. Postsynaptic events were evoked by optically stimulating channelrhodopsin expressing terminals from the BLA in the PrL cortex. This involved using Axograph to trigger a 5-ms pulse of light from an MDL-III-447 diode blue laser collimated to fit the microscope. The optical stimulation intensity was varied to establish the stimulus–response relationship. Once determined, the intensity was set either to induce a half-maximal response (for E/I balance experiments) or a maximal response (for AMPA/NMDA ratio experiments).

For experiments measuring the E/I balance of optically evoked postsynaptic currents onto PrL[PAG] neurons and PrL PVINs, neurons were held at –70 mV for recordings of oEPSCs and +10 mV for recordings of oIPSCs. The peak amplitudes of the oEPSC and oIPSC events were analyzed, and their ratio (oEPSC/oIPSC) was computed.

For experiments measuring the AMPA/NMDA ratio at BLA inputs onto PrL[PAG] neurons, 1 µM TTX and 100 µM 4-AP were included in the aCSF to isolate monosynaptic transmission (*Cho et al., 2013*; *Petreanu et al., 2009*). Neurons were held at +40 mV and optically evoked AMPA (oAMPA) and NMDA (oNMDA) currents were isolated using the following procedure: first, a combined oAMPA and oNMDA current was recorded in aCSF containing 100 µM picrotoxin. Subsequently, 50 µM dl-APV was added to the recording aCSF to isolate the oAMPA current. Finally, to isolate the oNMDA current, the oAMPA current was subtracted from the combined oAMPA and oNMDA current. The AMPA/

NMDA ratio was then computed based on the amplitude of each current. For PrL PVINs, the same procedure was followed with one additional step – after recording the AMPA current with the cell held at +40 mV, the holding potential was lowered to –70 mV and a final oAMPA current was recorded. The AMPA/NMDA ratio was then computed using the AMPA current recorded at –70 mV and the NMDA current recorded at +40 mV. This was done as PVINs express large numbers of calcium permeable AMPA receptors.

For experiments measuring optically evoked aEPSCs, TTX (1 µM) and 4-AP (100 µM) were added to the recording aCSF and $SrCl_2$ (2 mM) was substituted for $CaCl_2$. The substitution of strontium for calcium induces asynchronous neurotransmitter release after the initial release event. The resulting interevent interval and amplitude of the asynchronous events are commonly used to quantify pre- and postsynaptic function within defined circuits (*Choi and Lovinger, 1997*; *Dodge et al., 1969*; *Xu-Friedman and Regehr, 2000*). Recordings of aEPSCs were collected from neurons voltage clamped at –70 mV and analyzed within a 400-ms window beginning 50-ms poststimulation.

## Statistical analyses

Statistical analyses were performed using Stata 15.1 (StataCorp LLC, College Station, TX). Data were assessed for normality using the Wilks–Shapiro test and checked for outliers using the IQR rule. The experimental unit for this study was the individual animal. As such, while electrophysiological measures (e.g. E/I balance, intrinsic excitability) were obtained from multiple neurons within each animal, the data were averaged within each animal prior to analysis and reporting. Unless otherwise indicated, all data were analyzed using ANOVA models including all relevant factors. Repeated measures analyses used the Greenhouse–Geisser correction for sphericity. Post hoc tests were corrected for multiple comparisons using the Bonferroni method. All values reported are mean ± SEM. For purposes of statistical significance, $p < 0.05$ was considered significant.

## Acknowledgements

The authors would like to thank members of the Chandler Lab for their help carrying out the ethanol vapor exposure. This work was supported by NIH grants AA019967 (LJC), T32 AA007474 (JDO), and F32 AA030193 (JDO). The authors declare that the research was carried out in the absence of any commercial or financial relationships that could be construed as a conflict of interest. The artwork depicting electrophysiological recordings from PrL[PAG] neurons and PVINs in *Figures 4–12* was drawn in part using images from Servier Medical Art. Servier Medical Art is licensed under a Creative Commons Attribution 4.0 Unported License (https://creativecommons.org/licenses/by/4.0/). The depiction of a rat in *Figure 12* was obtained from scidraw.io and is licensed under a Creative Commons Attribution 4.0 Unported License (https://creativecommons.org/licenses/by/4.0/).

## Additional information

### Funding

| Funder | Grant reference number | Author |
| --- | --- | --- |
| National Institute on Alcohol Abuse and Alcoholism | AA019967 | L Judson Chandler |
| National Institute on Alcohol Abuse and Alcoholism | AA007474 | J Daniel Obray |
| National Institute on Alcohol Abuse and Alcoholism | AA030193 | J Daniel Obray |

The funders had no role in study design, data collection, and interpretation, or the decision to submit the work for publication.

## Author contributions

J Daniel Obray, Conceptualization, Formal analysis, Funding acquisition, Investigation, Visualization, Methodology, Writing – original draft, Project administration, Writing – review and editing; Erik T Wilkes, Investigation; Mike Scofield, Supervision, Writing – review and editing; L Judson Chandler, Conceptualization, Supervision, Funding acquisition, Writing – review and editing

## Author ORCIDs

J Daniel Obray ⓘ https://orcid.org/0000-0001-7526-4869
Mike Scofield ⓘ http://orcid.org/0000-0002-2330-6999
L Judson Chandler ⓘ https://orcid.org/0000-0002-1468-7320

## Ethics

All procedures were carried out in accordance with the National Research Council's Guide for the Care and Use of Laboratory Animals (2011) and were approved by the Medical University of South Carolina Institutional Animal Care and Use Committee (protocol # 2020-01161).

Reviewer #1 (Public review): https://doi.org/10.7554/eLife.101667.3.sa1
Reviewer #2 (Public review): https://doi.org/10.7554/eLife.101667.3.sa2
Reviewer #3 (Public review): https://doi.org/10.7554/eLife.101667.3.sa3
Author response https://doi.org/10.7554/eLife.101667.3.sa4

# Additional files

## Supplementary files

MDAR checklist

## Data availability

Raw data are presented as individual data points in figures where feasible. Numerical raw data is included in source data files and has also been published at https://doi.org/10.5061/dryad.wwpzgmswd.

The following dataset was generated:

| Author(s) | Year | Dataset title | Dataset URL | Database and Identifier |
|---|---|---|---|---|
| Obray JD, Wilkes ET, Scofield M, Chandler LJ | 2025 | Adolescent alcohol exposure, pain, and synaptic function at BLA inputs onto prelimbic neurons | https://doi.org/10.5061/dryad.wwpzgmswd | Dryad Digital Repository, 10.5061/dryad.wwpzgmswd |

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
