## [Editor Report · eLife Assessment]

This manuscript presents **important** information as to how adolescent alcohol exposure (AIE) alters pain behavior and relevant neurocircuits, with **convincing** data. The manuscript focuses on how AIE alters the basolateral amygdala, to the PFC (PV-interneurons), to the periaquaductal gray circuit, resulting in feed-forward inhibition. The manuscript is a detailed study of the role of alcohol exposure in regulating the circuit and reflexive pain, however, the role of the PV interneurons in mechanistically modulating this feed-forward circuit could be more strongly supported.

---

## [Referee Report · Reviewer #1 (Public review)]

Summary:

In this manuscript by Obray et al., the authors show that adolescent ethanol exposure increases mechanical allodynia in adulthood. Additionally, the show that BLA mediated inhibition of prelimbic cortex is reduced, resulting in increased excitability in neurons that then project to vlPAG. This effect was mediated by BLA inputs onto PV interneurons. The primary finding of the manuscript is that these AIE induced changes further impact acute pain processing in the BLA-PrL-vlPAG circuit, albeit behavioral readouts after inducing acute pain were not different between AIE rats and controls. These results provide novel insights into how AIE can have long lasting effects on pain-related behaviors and neurophysiology.In this manuscript by Obray et al., the authors show that adolescent ethanol exposure increases mechanical allodynia in adulthood. Additionally, the show that BLA mediated inhibition of prelimbic cortex is reduced, resulting in increased excitability in neurons that then project to vlPAG. This effect was mediated by BLA inputs onto PV interneurons. The primary finding of the manuscript is that these AIE induced changes further impact acute pain processing in the BLA-PrL-vlPAG circuit, albeit behavioral readouts after inducing acute pain were not different between AIE rats and controls. These results provide novel insights into how AIE can have long lasting effects on pain-related behaviors and neurophysiology.

The manuscript was very well written and the experiments were rigorously conducted. The inclusion of both behavioral and neurophysiological circuit recordings was appropriate and compelling. The authors analyzed their data extensively, and consider how many different factors may influence physiological activity and downstream behavior. The attention to SABV and appropriate controls was well thought out. The Discussion provided novel ideas for how to think about AIE and chronic pain, and proposed several interesting mechanisms. This was a very well executed set of experiments.

Comments on revisions:

The authors have addressed the concerns raised by the reviewers. Excellent work!

---

## [Referee Report · Reviewer #2 (Public review)]

Summary:

The study by Obray et al. entitled "Adolescent alcohol exposure promotes mechanical allodynia and alters synaptic function at inputs from the basolateral amygdala to the prelimbic cortex" investigated how adolescent intermittent ethanol exposure (AIE) affects the BLA -> PL circuit, with an emphasis on PAG projecting PL neurons, and how AIE changes mechanical and thermal nociception. The authors found that AIE increased mechanical, but not thermal nociception, and an injection of an inflammatory agent did not produce changes in an ethanol-dependent manner. Physiologically, a variety of AIE-specific effects were found in PL neuron firing at BLA synapses, suggestive of AIE-induced alterations in neurotransmission at BLA-PVIN synapses.

Strengths:

This was a comprehensive examination of the effects of AIE on this neural circuit, with an in-depth dissection of the various neuronal connections within the PL.

Sex was included as a biological variable, yet, there were little to no sex differences in AIE's effects, suggestive of similar adaptations in males and females.

Comments on revisions:

The authors addressed the reviews from the first submission which has substantially strengthened the conclusions of the study, including acknowledgement of unanswered questions for future studies to address.

---

## [Referee Report · Reviewer #3 (Public review)]

Summary:

Obray et al. investigate the long-lasting effects of adolescent intermittent ethanol (AIE) in rats, a model of alcohol dependence, on a neural circuit within prefrontal cortex. The studies are focused on inputs from the basolateral amygdala (BLA) onto parvalbumin (PV) interneurons and pyramidal cells that project to the periaqueductal gray (PAG). The authors found that AIE increased BLA excitatory drive onto parvalbumin interneurons and increased BLA feedforward inhibition onto PAG-projecting neurons.

Strengths:

Fully powered cohorts of male and female rodents are used, and the design incorporates both AIE and an acute pain model. The authors used several electrophysiological techniques to assess synaptic strength and excitability from a few complimentary angles. The design and statistical analysis are sound, and the evidence supporting synaptic changes following AIE results is convincing. The authors have also revised the Discussion to assimilate the findings within prior work out of their lab and others.

Weaknesses:

(1) There is incomplete evidence supporting some of the conclusions drawn in this manuscript. The authors claim the changes in feedforward inhibition onto pyramidal cells are due to the changes in parvalbumin interneurons; however, the authors did not determine that PV cells mediate the feedforward BLA op-IPSCs and changes following AIE (this would require a manipulation to reduce/block PV-IN activity). This limitation in results and interpretation is important because prior work shows BLA-PFC feedforward IPSCs can be driven by somatostatin cells. Cholecystokinin cells are also abundant basket cells in PFC and have been recently shown to mediate feedforward inhibition from thalamus and ventral hippocampus, so it's also possible that CCK cells are involved in the effects observed here

(2) The authors conclude that the changes in this circuit likely mediate long-lasting hyperalgesia, but this is not addressed experimentally. In some ways, the focused nature of the study is a benefit in this regard, as there is extensive prior literature linking this circuit with pain behaviors in alternative models (e.g., SNI), but it should be noted that these studies have not assessed hyperalgesia stemming from prior alcohol exposure. While the current studies do not include a causative behavioral manipulation, the strength of the association between BLA-PL-PAG function and hyperalgesia could be bolstered by with current data if there were relationships detected between electrophysiological properties and hyperalgesia.

(3) It should be noted that asEPSC frequency can also reflect changes in number of functional/detectable synapses. This measurement is also fairly susceptible to differences in inter-animal differences in ChR2 expression. There are other techniques for assessing presynaptic release probability (e.g., PPR, MK-801 sensitivity) that would improve the interpretation of these studies if that is intended to be a point of emphasis.

---

## [Author Response]

The following is the authors’ response to the original reviews

**Reviewer #1:**
Major Concerns/Public ReviewComment 1: There is a mild disconnect between behavioral readout (reflexive pain) and neural circuits of interest (emotional). Considering that this circuit is likely engaged in the aversiveness of pain, it would have been interesting to see how carrageenan and/or AIE impacted non-reflexive pain measures. Perhaps this would reveal a potentiated or dysregulated phenotype that matches the neurophysiological changes reported. However, this critique does not take away from the value of the paper or its conclusions.

We agree that including measures of non-reflexive pain would enhance future studies and potentially reveal a phenotype that is closely related to the observed changes in neurophysiology.

Minor Concerns/RecommendationsComment 1: There are a few minor grammatical errors in the text, mostly in the captions. A close read should be able to identify these errors.

We have fixed what grammatical errors we found.

**Reviewer #2**:Major Concerns/Public ReviewNo major concerns.Minor Concerns/RecommendationsComment 1: If pain sensitivity was assessed at 3 time points post carrageenan administration, why were these data averaged? Were there no differences between the time points? The data from the 3 time points should be presented, either in a figure, table, or supplementary materials.

We averaged the pain sensitivity data across the 3 time points following carrageenan administration because we were trying to present this data in a more concise manner. Pain sensitivity did change over time following carrageenan administration. We have now included the unaveraged data in figure 2 (panels D, F, H, and J).

Comment 2: For the optically-evoked EPSCs and IPSCs, were the peak amplitudes the max responses that could be obtained? If not, how were levels of ChR2 expression or light intensity controlled for?

The peak amplitudes for EPSCs and IPSCs were half the maximal response that could be evoked by optical stimulation. The AMPA and NMDA currents were maximal responses as prior literature indicated some PVINs have small NMDA currents, and we wanted to ensure these currents would be detected reliably. We updated our methods section to include this information in the voltage clamp recordings section.

Comment 3: In the example traces for the aEPSC experiment, the figure legend states that the "+" symbol indicates an asynchronous event. However, there are several "|" or "-" symbols in the figure. Perhaps this is an issue with the resolution of the figure and those are supposed to be "+"s.

We have increased the resolution of the figures to ensure that the markings of the asynchronous events display properly. We apologize for not noticing that these symbols were not displayed correctly in the original figures included in the manuscript.

Comment 4: For the von Frey and the Hargreaves test, were animals acclimated to the apparatus in the days leading up to the first test, or was the 5-minute pre-test the only acclimation that was done? This information needs to be provided. If the latter, there is concern that the animals did not fully acclimate to the apparatus and handling prior to testing, which should be taken into consideration in the interpretation of the behavioral analyses.

The rats underwent handling once a day for three days prior to the first von Frey and Hargreaves tests. On the day prior to the first test, rats were acclimated to the von Frey and Hargreaves apparatuses. The acclimation period consisted of a 15-min exposure to the von Frey apparatus and a 30-min exposure to the Hargreaves apparatus for each animal. This information has been added to the revised methods section under the assessment of mechanical and thermal sensitivity heading.

**Reviewer #3**:Major Concerns/Public ReviewComment 1: There is incomplete evidence supporting some of the conclusions drawn in this manuscript. The authors claim that the changes in feedforward inhibition onto pyramidal cells are due to the changes in parvalbumin interneurons, but evidence is not provided to support that idea. PV cells do not spontaneously fire action potentials spontaneously in slices (nor do they receive high levels of BLA activity while at rest in slices). It is possible that spontaneous GABA release from PV cells is increased after AIE but the authors did not report sIPSC frequency. Second, the authors did not determine that PV cells mediate the feedforward BLA op-IPSCs and changes following AIE (this would require manipulation to reduce/block PV-IN activity). This limitation in results and interpretation is important because prior work shows BLA-PFC feedforward IPSCs can be driven by somatostatin cells. Cholecystokinin cells are also abundant basket cells in PFC and have been recently shown to mediate feedforward inhibition from the thalamus and ventral hippocampus, so it's also possible that CCK cells are involved in the effects observed here.

The hypothesis that adolescent alcohol exposure could change spontaneous GABA release from PVINs is an interesting one that merits future exploration. Unfortunately, as the focus of this manuscript was on circuit-specific alterations in synaptic function, this experiment is somewhat outside the scope of the paper as sIPSCs and mIPSCs are not circuit specific measures of GABA activity and would not reflect spontaneous release from only GABA interneurons receiving input from the BLA. Despite this, a future study investigating spontaneous GABA release from PVINs in the PrL would be a valuable complement to the present study.

While we did not directly manipulate PVINs to demonstrate that decreased oIPSC amplitude at PrL^PAG^ neurons following AIE is due solely to changes in PVINs, it is notable that both the intrinsic excitability of PVINs and the BLA-driven E/I balance at PVINs were reduced following AIE. These changes would be consistent with decreased PVIN output onto PrL^PAG^ neurons. However, we agree that this does not preclude the possibility that changes in SST or CCK interneurons contribute to the observed decrease in BLA-driven inhibition at PrL^PAG^ neurons following AIE. As such, we have altered the wording in the discussion to indicate that reduced BLA-driven feedforward inhibition of PrL^PAG^ neurons may be related, at least in part, to the observed changes in PVINs.

Comment 2: The authors conclude that the changes in this circuit likely mediate long-lasting hyperalgesia, but this is not addressed experimentally. In some ways, the focused nature of the study is a benefit in this regard, as there is extensive prior literature linking this circuit with pain behaviors in alternative models (e.g., SNI), but it should be noted that these studies have not assessed hyperalgesia stemming from prior alcohol exposure. While the current studies do not include a causative behavioral manipulation, the strength of the association between BLA-PL-PAG function and hyperalgesia could be bolstered by current data if there were relationships detected between electrophysiological properties and hyperalgesia. Have the authors assessed this? In addition, this study is limited by not addressing the specificity of synaptic adaptations to the BLA-PL-PAG circuit. For instance, PL neurons send reciprocal projections to BLA and send direct projections to the locus coeruleus (which the authors note is an important downstream node of the PAG for regulating pain).

We have not assessed correlations between the electrophysiological properties and hyperalgesia. We feel that future studies using DREADDs to perform cell-type and circuit-specific manipulations can better address the involvement of this circuitry in long-lasting hyperalgesia following AIE. With respect to the circuit specificity of the observed changes, we have previously evaluated the effects of AIE on pyramidal neurons projecting from the PrL to the BLA (PrL^BLA^). We found that following AIE exposure there was no change in the intrinsic excitability of these neurons. In addition, the amplitude and frequency of sEPSCs and sIPSCs onto PrL^BLA^ neurons was unchanged. While these results did not assess whether the BLA-PrL-BLA circuit undergoes synaptic adaptations similar to those observed in the BLA-PrL-vlPAG circuit, it is notable that the intrinsic excitability of PrL^BLA^ neurons was unchanged following AIE exposure. This indicates that the effects of AIE on the intrinsic excitability of pyramidal neurons in the PrL may be circuit specific. We agree that it would be interesting to study the effect of AIE on PrL neurons that project to the locus coeruleus, however due to the well-defined role of the BLA-PrL-vlPAG circuit in pain we chose to evaluate this circuit first.

Comment 3: I have some concerns about methodology. First, 5-ms is a long light pulse for optogenetics and might induce action-potential independent release. Does TTX alone block op-EPSCs under these conditions? Second, PV cells express a high degree of calcium-permeable AMPA receptors, which display inward rectification at positive holding potentials due to blockade from intracellular polyamines. Typically, this is controlled/promoted by including spermine in the internal solution, but I do not believe the authors did that. Nonetheless, the relatively low A/N ratios for this cell type suggest that CP-AMPA receptors were not sampled with the +40/+40 design of this experiment, raising concerns that the majority of AMPA receptors in these cells were not sampled during this experiment. Finally, it should be noted that asEPSC frequency can also reflect changes in a number of functional/detectable synapses. This measurement is also fairly susceptible to differences in inter-animal differences in ChR2 expression. There are other techniques for assessing presynaptic release probability (e.g., PPR, MK-801 sensitivity) that would improve the interpretation of these studies if that is intended to be a point of emphasis.

When we included TTX but not 4-AP we did not observe any optically evoked responses, so we don’t believe that the 5-ms pulse induced action-potential independent release in these experiments. With respect to the second point, we did not include spermine in the internal solution for the AMPA/NMDA recordings in PVINs, and it is possible that endogenous polyamines interfered with recording CP-AMPA receptors in the +40/+40 design. To address this concern, we recalculated the AMPA/NMDA ratio for PVINs using data from an optically evoked AMPA current that was collected while holding the cell at -70 mV. This data was collected at the end of the +40/+40 recording protocol as we were interested in assessing whether there would be any difference in the ratio of the +40/-70 AMPA current across treatment conditions. As there were no observed difference in the +40/-70 AMPA current ratio across treatment groups, we had originally used the +40 AMPA current for calculating the AMPA/NMDA ratio for PVINs to make the methods for calculating this ratio uniform for both PVINs and PrL^PAG^ neurons. The methods, results, and Fig. 10 have been updated to reflect the recalculated AMPA/NMDA ratio for PVINs. Notably, only the significance of the AIE x carrageenan interaction was altered by the change in the way the AMPA/NMDA ratio was calculated. Originally, this interaction displayed a trend toward significance (*p* = 0.0501), however when the recalculated AMPA/NMDA ratio was analyzed this interaction term became significant (*p* = 0.0131). We have also added the +40/-70 AMPA ratio to figure 10 as it might be of interest.

Finally, the point regarding aEPSC frequency reflecting not only release probability but also the number of functional/detectable synapses is an important consideration. For this manuscript, we intentionally selected aEPSC frequency for this reason. As the BLA to PrL projection continues to mature during adolescence, the number of BLA contacts onto GABA neurons in the PrL increases. Thus, we thought that it was possible that AIE would alter the number of detectable BLA inputs onto PVINs. We acknowledge that as this measure is sensitive to differences in ChR2 expression between animals/slices it can be difficult to interpret. We also agree that in the future it would be beneficial to include either PPR or MK-801 sensitivity to improve interpretability.

Comment 4: In a few places in the manuscript, results following voluntary drinking experiments (especially Salling et al. and Sicher et al.) are discussed without clear distinction from prior work in vapor models of dependence.

We have altered the manuscript to specifically note where voluntary drinking was used rather than vapor models.

Comment 5: Discussion (lines 416-420). The authors describe some differing results with the literature and mention that the maximum current injection might be a factor. To me, this does not seem like the most important factor and potentially undercuts the relevance of the findings. Are the cells undergoing a depolarization block? Did the authors observe any changes in the rheobase or AP threshold? On the other hand, a more likely difference between this and previous work is that the proportion of PAG-projecting cells is relatively low, so previous work in L5 likely sampled many types of pyramidal cells that project to other areas. This is a key example where additional studies by the current group assessing a distinct or parallel set of pyramidal cells would aid in the interpretation of these results and help to place them within the existing literature. Along these lines, PAG-projecting neurons are Type A cells with significant hyperpolarization sag. Previous studies showed that adolescent binge drinking stunts the development of HCN channel function and ensuing hyperpolarization sag. Have the authors observed this in PAG-projecting cells? Another interesting membrane property worth exploring with the existing data set is the afterhyperpolarization / SK channel function.

In discussing the maximum current injection as a factor in differing results on intrinsic excitability, we were principally considering how the additional data points increase the power of the analysis and thus the likelihood of detecting an effect. In focusing on this, however, we ignored other relevant and interesting factors that we should also have discussed. Additional analyses examining HCN and SK channel function have now been added to the manuscript and incorporated into the results section under the heading Adolescent Intermittent Ethanol Exposure and Carrageenan Enhanced the Intrinsic Excitability of Prelimbic Neurons Projecting to the Ventrolateral Periaqueductal Gray. We have also modified the third paragraph in the discussion to add additional context. Additional information on the biophysical properties of the neurons has been added to Figure 4.

Minor Concerns/RecommendationsComment 1: Subheadings are vague. "Analysis of..." Should be rephrased to use active voice to describe key findings.

The subheadings have been rephrased to describe key findings.

Comment 2: Consider altering or consolidating the figure layout for clarity. For instance, it would be helpful for aEPSCs to be near the AMPA and NMDA experiments. The feedforward IPSCs could also be with the PV-IN recordings. This would be helpful in developing a cohesive picture of key findings. To that end, a working model or graphical abstract would be helpful.

It doesn’t appear that this journal allows graphical abstracts, but we have added a model that summarizes the principal findings in the discussion.

Comment 3: There are a lot of statistics punctuating the text in the Results. It can be hard to parse at times.

We considered moving the statistics to tables, but this became unwieldy.

Comment 4: The Discussion is quite long (10 paragraphs). Suggest consolidating to 3-4 most salient points.

We appreciate this comment and have made some edits to the discussion, albeit without consolidating it to only 3-4 points.